# Fair and Efficient Allocations Without Obvious Manipulations

**Alexandros Psomas**
Purdue University
apsomas@cs.purdue.edu

**Paritosh Verma**
Purdue University
verma136@purdue.edu

## Abstract

We consider the fundamental problem of allocating a set of indivisible goods among strategic agents with additive valuation functions. It is well known that, in the absence of monetary transfers, Pareto efficient and truthful rules are dictatorial, while there is no deterministic truthful mechanism that allocates all items and achieves envy-freeness up to one item (EF1), even for the case of two agents. In this paper, we investigate the interplay of fairness and efficiency under a relaxation of truthfulness called non-obvious manipulability (NOM), recently proposed by [TM20]. We show that this relaxation allows us to bypass the aforementioned negative results in a very strong sense. Specifically, we prove that there are deterministic and EF1 algorithms that are not obviously manipulable, and the algorithm that maximizes utilitarian social welfare (the sum of agents' utilities), which is Pareto efficient but not dictatorial, is not obviously manipulable for $n \geq 3$ agents (but obviously manipulable for $n = 2$ agents). At the same time, maximizing the egalitarian social welfare (the minimum of agents' utilities) or the Nash social welfare (the product of agents' utilities) is obviously manipulable for any number of agents and items. Our main result is an approximation preserving black-box reduction from the problem of designing EF1 and NOM mechanisms to the problem of designing EF1 algorithms. En route, we prove an interesting structural result about EF1 allocations, as well as new "best-of-both-worlds" results (for the problem without incentives), that might be of independent interest.

## 1 Introduction

We consider the fundamental problem of allocating a set of indivisible items among strategic agents with additive preferences. It is well-understood that, in the absence of monetary transfers, fairness, efficiency and truthfulness cannot be reconciled, in a very strong sense. For example, a serial dictatorship (arguably the most unfair rule) is the *unique* truthful and Pareto efficient mechanism [KM02, EK03], even for the case of two agents and randomized mechanisms (or, equivalently, divisible items) [Sch96]. At the same time, achieving envy-freeness up to one item (EF1) [Bud11], a popular fairness notion for indivisible goods that is compatible with Pareto efficiency [CKM$^+$19], is impossible for truthful and deterministic mechanisms to achieve, even for two additive agents [ABCM17].

The standard abstraction for a strategic agent assumes that, when presented with an algorithm, the agent will perfectly understand it and optimally respond to it. Therefore, a truthful mechanism should protect against all deviating strategies. This modeling is overly pessimistic since, for example, in practice agents do not have exact knowledge about others' preferences. Since truthfulness rules out reasonable solutions, previous work typically ignores strategic issues altogether. In this paper we aim to escape the aforementioned impossibility results by relaxing truthfulness: our goal is to design mechanisms that protect only against *obviously dominant* deviating strategies.

36th Conference on Neural Information Processing Systems (NeurIPS 2022).

Informally, a strategy is obviously dominant if it guarantees an outcome better than any outcome of every other strategy [Li17]. Designing obviously truthful mechanisms, those where honest reporting is obviously dominant, is a problem that has attracted significant attention in recent years [AG18, KV19, FV19, AMN20]. Since dishonesty is also a strategy, a deviation can be profitable but not obviously profitable. A manipulation is obvious if it yields a higher utility than truth-telling in either the best or worst-case scenarios. Seminal work of [TM20] defines this notion, and shows that relaxing truthfulness to non-obvious manipulability can strictly improve the designer's objective in a number of domains (e.g. two-sided matching). In this paper we aim to provide a comprehensive study of what can and what cannot be achieved by non-obviously manipulable mechanisms in the fundamental domain of fair and efficient allocation of indivisible goods.

**Our Contribution.** We first consider the problem of designing a non-obviously manipulable mechanism (NOM) which always outputs allocations that are envy-free up to one item (EF1), a task that is impossible for truthful mechanisms to achieve [ABCM17]. We prove that the Round-Robin procedure (agents choose items one at a time, following a fixed order) which is known to always output EF1 allocations, is also not obviously manipulable (Theorem 1), giving a separation between truthfulness and non-obviously manipulability for our problem. In fact, we show that NOM is compatible with even stronger notions of fairness. Specifically, a randomized algorithm can be ex-post EF1 (i.e. the output allocation is always EF1), and simultaneously be envy-free in expectation (ex-ante EF). There are known algorithms that satisfy this "best-of-both-worlds" guarantee [FSV20, Azi20b]; we show that the PS-Lottery algorithm of [Azi20b] is not-obviously manipulable. This result uses a connection between ex-ante proportionality and not obvious manipulability; we later exploit this connection to establish new negative "best-of-both-worlds" results.

We proceed to study efficient algorithms. We focus on the three most prominent notions of efficiency: utilitarian social welfare (the sum of agents' utilities), egalitarian social welfare (the minimum of agents' utilities), and Nash social welfare (the geometric mean of agents' utilities). For the case of $n = 2$ agents all three notions are incompatible with non obvious manipulability, i.e. every (deterministic or randomized) algorithm that always outputs an integral solution that is optimal with respect to any of these objectives is obviously manipulable (Theorem 2). This is also true for the case of more than three agents for egalitarian and Nash social welfare (Theorems 8 and 4). The high level intuition for these impossibilities is as follows. There are instances where a specific agent[1] gets only her least favorite item, in the worst-case. However, maximizing egalitarian or Nash social welfare ensures that the number of agents that get non-zero utility are maximized. Therefore, by reporting that she only values a single item (her favorite item) to the mechanism, this agent can force the mechanism to give her this item (or increase the probability that she gets it) in the worst-case. Note that lack of agents' knowledge about each others' preferences is used as a justification for why manipulations might not be a first-order concern in our domain [CKM+19]; the fact that maximizing Nash social welfare is manipulable even when agents know nothing about each other further highlights the need for more nuanced formal models between "agents are always honest" and "agents are perfectly rational and all-knowing expected utility maximizers." Surprisingly, the same is not true for utilitarian social welfare: there is a NOM mechanism that always outputs an allocation that maximizes utilitarian social welfare for the case of $n \geq 3$ agents (Theorem 3). This gives an efficient and not obviously manipulable mechanism that is not dictatorial.

Since maximizing Nash social welfare, an objective that simultaneously achieves Pareto efficiency and EF1, is obviously manipulable, the next natural question is whether there are any other Pareto efficient and EF1 algorithms that satisfy NOM. Our main result answers this question in the affirmative. In fact, we prove that the problem of designing a Pareto efficient, EF1 and NOM *mechanism* is exactly as hard as designing a Pareto efficient and EF1 *algorithm*. We give a black-box reduction, that preserves Pareto efficiency guarantees, which given an algorithm that always outputs (clean and non-wasteful) EF1 allocations, outputs a mechanism that is not obviously manipulable and always outputs EF1 allocations. There are two crucial steps in our reduction. Given the valuation function $\mathbf{v}_i$ of agent $i$, let $\mathrm{EF1}(i, \mathbf{v}_i)$ be the set of allocations that are clean, non-wasteful and EF1 for agent $i$. First, our reduction ensures that, when agent $i$ reports her valuation to be $\mathbf{b}_i$, every single allocation in $\mathrm{EF1}(i, \mathbf{b}_i)$ is possible, i.e. for every allocation $A \in \mathrm{EF1}(i, \mathbf{b}_i)$ there are reports for the other agents such that $A$ is the output of the mechanism. Second, we prove the following intuitive structural result, that might be of independent interest. For an agent $i$ having valuation function $\mathbf{v}_i$, the worst-case allocation in $\mathrm{EF1}(i, \mathbf{v}_i)$ is better than the worst-case allocation in $\mathrm{EF1}(i, \mathbf{b}_i)$ (Lemma 4). These facts

---

[1]Informally, the agent that the tie-breaking rule favors.

Table 1: Existence of NOM mechanisms for various fairness and efficiency notions.

| Fairness & Efficiency Notions | Existence of NOM Mechanism |
|---|---|
| EF1 | ✓ [Theorem 1] |
| Ex-ante EF & Ex-post EF1 | ✓ [Theorem 7] |
| Utilitarian Social Welfare | ✗ $n = 2$ agents [Theorem 2] 
 ✓ $n \geq 3$ agents [Theorem 3] |
| Egalitarian Social Welfare | ✗ [Theorem 8] |
| Nash Social Welfare | ✗ [Theorem 4] |
| fPO + EF1 | ✓ [Theorem 5] |
| Ex-ante EF & Ex-post fPO + EF1 | ✗ [FSV20] |

combined establish the "worst-case" part from the definition of NOM, which is the most challenging step. As a direct application of our reduction, by giving as input the algorithmic results [GM21], we get a fractionally Pareto efficient, EF1 and NOM mechanism for additive agents; this last result cannot be improved by adding ex-ante fairness guarantees (due to a theorem of [FSV20]). Finally, despite the aforementioned technical subtleties, the "code" of our reduction is fairly simple: it checks whether some allocations are EF1, and if not it calls the black-box algorithm. This simplicity is a feature, with concrete practical implications: by adding a few lines of code to the implementation of any EF1 and Pareto efficient algorithm (e.g. the algorithm that maximizes Nash social welfare, which is used by the popular website Spliddit [GP15]) one can maintain these guarantees, while provably protecting against certain types of manipulations.

Our results show a connection between certain notions of fairness and not obvious manipulability: positive algorithmic results for EF1 and "best-of-both-worlds" guarantees can be used to get positive results for not obviously manipulable mechanisms. In Appendix E we exploit this connection in the other direction, and show how negative results for not obviously manipulable mechanisms can be used to prove new negative results for "best-of-both-worlds" algorithms. Specifically, we prove that it is impossible to achieve ex-ante proportionality, ex-post Pareto efficiency while ex-post maximizing the number of agents with positive utility. As direct corollaries we recover a known result of [FSV20] that ex-ante proportional and ex-post MNW allocations do not exist, as well as new results: it is impossible to achieve ex-ante proportionality by randomizing over allocations that are (1) ex-post Pareto efficient and ex-post egalitarian, or (2) ex-post leximin.

**Related work.** [TM20] define the notion of not obvious manipulability. They show that a number of known mechanisms that are not truthful satisfy not obviously manipulability, e.g. stable mechanisms in the context of two-sided matching are not obviously manipulable (in direct contrast to [Rot82], who shows that there exists no mechanism that is both stable and truthful). At the same time, many known mechanisms that are not truthful are obviously manipulable, e.g. the first-price auction, and more generally the pay-as-bid auction.

The notion of non obvious manipulability has been explored in some very recent works [OSH19, AL21]. [OSH19] study cake-cutting and show that, as opposed to truthfulness, NOM is compatible with proportionality: an adaptation of the well-known moving knife procedure satisfies both properties. They also observe that every proportional (direct revelation) rule satisfies the worst-case guarantee required for NOM (we show a similar statement in Lemma 5), while every Pareto efficient (direct revelation) rule satisfies the best-case guarantee required for NOM. [AL21] explore obvious manipulations in voting theory. They give sufficient conditions for voting rules to be not obviously manipulable, and show that a number of voting rules escape the pessimistic conclusions of the Gibbard-Satterthwaite theorem, e.g., the Borda rule is not obviously manipulable.

Other than relaxing the notion of incentive compatibility, one can escape the aforementioned impossibility results by restricting agents' valuations, e.g. focus on dichotomous [HPPS20, BEF21, BCIZ21, BV21] or Leontief valuations [GZH+11, FGP14, PPS15], or by using money-burning (wasting resources) as a substitute for payments [HR08, CGG13, FTTZ16, FGPS19, ACGH20].

## 2 Preliminaries

We consider the problem of allocating a set M of $m$ items among a set N of $n$ agents with additive utilities. We use $[k]$ to denote the set $\{1, 2, \ldots, k\}$ for any positive integer $k \in \mathbb{Z}_+$. Proofs pertaining to randomized mechanisms are deferred to the appendix; see Appendix A for the corresponding preliminaries on randomized allocations/mechanisms.

**Allocations.** A *fractional allocation* $A \in [0,1]^{n \cdot m}$ is a $m \times n$ matrix that defines for each agent $i \in \mathbb{N}$ and item $j \in \mathbb{M}$ the fraction $A_{i,j}$ of the item $j$ that the agent $i$ receives. We represent a fractional allocation as $A = (A_1, \ldots, A_n)$ where $A_i = (A_{i,1}, \ldots, A_{i,m}) \in [0,1]^m$ denotes the fractions of all items allocated to agent $i$. A feasible allocation satisfies $\sum_{i \in \mathbb{N}} A_{i,j} \leq 1$, for all $j \in \mathbb{M}$. A fractional allocation $A$ is *integral* if $A_i \in \{0,1\}^m$ for all agents $i \in \mathbb{N}$. An integral allocation can be equivalently defined as $n$ disjoint subsets of the set of items M, this representation is often convenient. Let $\Pi_n(\mathbb{M}) = \{(S_1, \ldots, S_n) \mid \cup_{i=1}^n S_i \subseteq \mathbb{M} \text{ and } \forall i \neq j, \ S_i \cap S_j = \emptyset\}$ denote the set of all $n$ ordered disjoint subsets of the set of items M. Given an integral allocation $A = (A_1, \ldots, A_n)$, we can interpret the binary vectors $A_i = (A_{i,1}, \ldots, A_{i,m})$ as subsets of items $A_i := \{j \in \mathbb{M} \mid A_{i,j} = 1\}$. This allows us to view an integral allocation $A$ as $n$ ordered disjoint subsets of items, i.e., $A \in \Pi_n(\mathbb{M})$. An integral allocation $A$ is *complete* if $\cup_{i=1}^n A_i = M$ and *partial* if $\cup_{i=1}^n A_i \subset M$. Unless stated otherwise, we will use the term allocation to refer to complete allocations. We will use the term *bundle* to refer to any subset of items.

**Preferences.** Each agent $i \in \mathbb{N}$ has a private valuation function $\mathbf{v}_i(.)$ that outputs the utility that agent $i$ derives from a given set (or fractions) of items. We use the notation $\mathbf{v}_i(X_i)$ (resp. $\mathbf{v}_i(A_i)$) to denote the utility that agent $i$ gets from the items allocated to her in a fractional allocation $X$ (resp. integral allocation $A$). We consider agents with additive utilities. An additive agent $i \in \mathbb{N}$ has a non-negative valuation $v_{i,j}$ for receiving the entirety of item $j$; her utility for an allocation $X$ is $\mathbf{v}_i(X_i) = \sum_{j \in \mathbb{M}} X_{i,j} v_{i,j}$; for an integral allocation $A$, the utility is simply $\mathbf{v}_i(A_i) = \sum_{j \in A_i} v_{i,j}$.

**Mechanisms.** A mechanism $\mathcal{M}$ elicits "bids" (i.e. reported valuations) $\mathbf{b} = (\mathbf{b}_1, \ldots, \mathbf{b}_n)$ from every agent $i \in \mathbb{N}$, and outputs a feasible allocation. We write $X_{i,j}(\mathbf{b})$ for the fraction of item $j$ allocated to agent $i$ when each agent $j \in \mathbb{N}$ reports a valuation $\mathbf{b}_j$. We use the notation $\mathbf{b}$ (and $\mathbf{b}_i$) to refer to the input to a mechanism, and $\mathbf{v}$ (and $\mathbf{v}_i$) to refer to the true valuations of agents. A mechanism $\mathcal{M}$ is *deterministic* if for every reported valuations $\mathbf{b}$ it deterministically outputs an integral allocation. We will use $\mathcal{M}(\mathbf{b}) = (\mathcal{M}_1(\mathbf{b}), \mathcal{M}_2(\mathbf{b}), \ldots, \mathcal{M}_n(\mathbf{b}))$ to denote the integral allocation that a mechanism $\mathcal{M}$ outputs given reported valuations $\mathbf{b} = (\mathbf{b}_1, \ldots, \mathbf{b}_n)$ as input, here $\mathcal{M}_i(\mathbf{b}) \subseteq \mathbb{M}$ represents the bundle of items that agent $i$ receives in the allocation $\mathcal{M}(\mathbf{b})$. For a deterministic mechanism $\mathcal{M}$, the value $\mathbf{v}_i(\mathcal{M}_i(\mathbf{b}))$ denotes the utility of agent $i$ for the allocation output by $\mathcal{M}$ on input $\mathbf{b}$.

**Notions of incentive compatibility.** A mechanism $\mathcal{M}$ is *truthful* if agents cannot strictly improve their utility by misreporting their valuation, i.e. for all $i \in \mathbb{N}$, valuations $\mathbf{v}_i, \mathbf{b}_i$, and reports of the other agents $\mathbf{v}_{-i}$, $\mathbf{v}_i(\mathcal{M}_i(\mathbf{v}_i, \mathbf{v}_{-i})) \geq \mathbf{v}_i(\mathcal{M}_i(\mathbf{b}_i, \mathbf{v}_{-i}))$. Our work focuses on a notion of incentive compatibility that is a relaxation of truthfulness called *not obvious manipulability*.

**Definition 1** (Not Obviously Manipulable [TM20]). A mechanism $\mathcal{M}$ is *not obviously manipulable* (NOM) if for every agent $i \in \mathbb{N}$ with valuation function $\mathbf{v}_i$, and every possible report $\mathbf{b}_i$ of agent $i$, the following two inequalities hold:

(1) $\min_{\mathbf{v}_{-i}} \mathbf{v}_i(\mathcal{M}_i(\mathbf{v}_i, \mathbf{v}_{-i})) \geq \min_{\mathbf{v}_{-i}} \mathbf{v}_i(\mathcal{M}_i(\mathbf{b}_i, \mathbf{v}_{-i}))$.

(2) $\max_{\mathbf{v}_{-i}} \mathbf{v}_i(\mathcal{M}_i(\mathbf{v}_i, \mathbf{v}_{-i})) \geq \max_{\mathbf{v}_{-i}} \mathbf{v}_i(\mathcal{M}_i(\mathbf{b}_i, \mathbf{v}_{-i}))$.

Intuitively, if a mechanism is NOM then an agent cannot increase her worst-case utility or her best-case utility (computed with respect to the true valuation) by misreporting her valuation. If either the worst-case or the best-case utility can be improved then the mechanism is *obviously manipulable*.

**Notions of efficiency.** An integral allocation $A = (A_1, \ldots, A_n)$ is *Pareto efficient* (or PO) iff there is no integral allocation $A' = (A'_1, \ldots, A'_n)$ such that for all agents $i \in \mathbb{N}$, $\mathbf{v}_i(A'_i) \geq \mathbf{v}_i(A_i)$, and for at least one agent this inequality is strict. An (integral or fractional) allocation $X = (X_1, \ldots, X_n)$ is *fractionally Pareto efficient* (or fPO) iff there is no fractional allocation $X' = (X'_1, \ldots, X'_n)$ such that for all agents $i \in \mathbb{N}$, $\mathbf{v}_i(X'_i) \geq \mathbf{v}_i(X_i)$, and for at least one agent this inequality is strict. Note that fractional Pareto efficiency is a strictly stronger notion than Pareto efficiency. An (integral or fractional) allocation $X$ is $\alpha$-approximately (resp. fractionally) Pareto efficient if there is no integral

allocation (resp. fractional allocation) $X' = (X_1', X_2', \ldots, X_n')$ such that $\alpha \cdot \mathbf{v}_i(X_i') \geq \mathbf{v}_i(X_i)$, with at least one of these inequalities strict [RF90, ILWM17, FGPS19, ZP20]. Notice that for $\alpha = 1$ we exactly recover Pareto efficiency (resp. fractional Pareto efficiency).

A (partial) allocation $A = (A_1, A_2, \ldots, A_n)$ is *non-wasteful* iff for each $i \in \mathrm{N}$, $v_{i,j} = 0$ for every unallocated item $j \in \mathrm{M} \setminus \cup_{k \in \mathrm{N}} A_k$[2]. A bundle $S \subseteq \mathrm{M}$ is *clean* for wrt valuation $\mathbf{v}_i$ if the items comprising $S$ have positive value, i.e., $\mathbf{v}_i(g) > 0$ for all $g \in S$. Further, a (partial) allocation $A = (A_1, A_2, \ldots, A_n)$ is clean if for each $i \in \mathrm{N}$ the bundle $A_i$ is clean wrt valuation $\mathbf{v}_i$.

Often, we are interested in computing or approximating specific points (i.e., allocations) present on the Pareto frontier that have additional desirable properties. The *utilitarian social welfare* of an allocation $X$, denoted by $\mathrm{SW}(X)$, is defined as the sum of utilities that each agent gets in the allocation $X$, i.e., $\mathrm{SW}(X) = \sum_{i \in \mathrm{N}} \mathbf{v}_i(X_i)$. The *Nash social welfare* of an (integral or fractional) allocation $X$, denoted by $\mathrm{NSW}(X)$, is defined as the geometric mean of the utilities of agents in the allocation $X$, i.e., $\mathrm{NSW}(X) = (\prod_{i \in \mathrm{N}} \mathbf{v}_i(x_i))^{\frac{1}{n}}$. An *integral* allocation that maximizes the Nash social welfare, among all integral allocations, is called a *Nash social welfare maximizing* (or MNW) allocation. There are instances where every integral allocation $A$ is such that $\mathrm{NSW}(A) = 0$, i.e., there is always an agent having zero utility. To cover such an edge case, integral MNW allocations are defined as follows [CKM$^+$19]. An integral allocation is MNW iff $(i)$ it maximizes, among the set of all integral allocations, the number of agents having positive utility and $(ii)$ for any such maximal set of agents $S$, it maximizes the geometric mean of the utilities of agents in $S$.

**Notions of fairness.** An (integral or fractional) allocation $X$ is called *envy-free* (EF) if for every pair of agents $i, j \in \mathrm{N}$, agent $i$ values her allocation at least as much as the allocation of agent $j$, i.e., $\mathbf{v}_i(X_i) \geq \mathbf{v}_i(X_j)$. Achieving envy-freeness is impossible for integral allocations (consider the case of a single item and two agents that both want it), so relaxation of envy-freeness are considered. An integral allocation $A$ is envy-free up to one item (EF1) if for every pair of agents $i, j \in \mathrm{N}$, where $A_j \neq \emptyset$, agent $i$ values her allocation at least as much as the allocation of agent $j$, subject to the removal of one item from agent $j$'s bundle, i.e., $\mathbf{v}_i(A_i) \geq \mathbf{v}_i(A_j \setminus \{g\})$ for some item $g \in A_j$.

Finally, we are often interested in, so called, best-of-both-worlds guarantees. Let $\mathcal{P}$ be a fairness or efficiency notion for integral allocations and $\mathcal{Q}$ be a fairness or efficiency notion for fractional allocations. A randomized allocation $\mathbf{R}$ (see Appendix A for definition), which outputs an integral allocation $A^z$ with probability $p^z$, satisfies the notion $\mathcal{P}$ *ex-post* if each integral allocation $A^z$ in the support of $\mathbf{R}$ satisfies $\mathcal{P}$. Additionally, the randomized allocation $\mathbf{R}$ satisfies the notion $\mathcal{Q}$ *ex-ante* if the expected fractional allocation $Y = \sum_{z=1}^{k} p^z \cdot A^z$ corresponding to $\mathbf{R}$ satisfies the notion $\mathcal{Q}$.

## 3  Fair Mechanisms

In this section we study whether non obvious manipulability is compatible with EF1. For deterministic mechanisms, it is known that no truthful mechanism can always outputs an EF1 allocation, even for the case of two agents [ABCM17]. In sharp contrast, we show that Round-Robin, arguably the simplest EF1 algorithm, is not obviously manipulable. Recall that the Round-Robin algorithm allocates items to agents over a sequence of rounds. In each round, the agents choose one item each (a highest-value remaining item, as per their valuation[3]) following a fixed, arbitrary order.

**Theorem 1.** *Round-Robin is* not obviously manipulable.

*Proof.* We prove that the two inequalities in Definition 1 hold for Round-Robin. Let $i \in \mathrm{N}$ be the $i$-th agent in the Round-Robin order, and let $\mathbf{v}_i$ be her valuation vector. Assume without loss of generality that $v_{i,j} \geq v_{i,j+1}$ for all $j = 0, \ldots, m - 1$. Let $\ell$ be the number of items agent $i$ receives (either $\lfloor \frac{n}{m} \rfloor$ or $\lfloor \frac{n}{m} \rfloor + 1$, depending on $i$, $n$ and $m$), and notice that $\ell$ does not depend on her report.

We first prove inequality (1) from the definition of NOM, i.e. the worst-case guarantee. The $k$-th time agent $i$ gets to pick an item, there is an un-allocated item that she values at least $v_{i,i+(k-1)n}$, since only $(k-1)n + i - 1$ items have been allocated at that point. Therefore, the worst-case for agent $i$ under honest reporting is realized when all other agents rank the items in the same order as her, and her utility is exactly $\sum_{k=1}^{\ell} v_{i,i+(k-1)n}$. Now consider the case when all agents other than $i$

---

[2]Note that, every Pareto efficient (fractionally Pareto efficient) integral allocation is non-wasteful.

[3]Ties can be broken in an arbitrary manner.

rank items in the same order as $i$ (i.e. $v_{i',j} \geq v_{i',j+1}$ for all $i' \in N$), and $i$ reports some vector $\mathbf{b}_i$. Let $j_1, \ldots, j_\ell$ be the items she receives, in the order she picked them. First, without loss of generality, $i$ picks these items in decreasing order with respect to her true valuation, i.e. $v_{i,j_k} \geq v_{i,j_{k+1}}$: if she picks an item $k$ at round $t$, and at a later round $t'$ she could pick an item $k'$ with $v_{i,k'} > v_{i,k}$, then by the choice of valuation for agents other than $i$, picking $k'$ in round $t$ and $k$ in round $t'$ is also possible. Second, $v_{i,j_k} \leq v_{i,i+(k-1)n}$: when $i$ picks an item for the $k$-th time, $(i-1) + (k-1)(n-1)$ of her top $i + (k-1)n$ items have been picked by other agents, and $k-1$ of her top $i+(k-1)n$ items have been picked by herself (since items are picked in decreasing order with respect to the true valuation). Therefore $v_{i,i+(k-1)n}$ is the largest value $v_{i,j_k}$ can have; $i$'s utility is at most $\sum_{k=1}^{\ell} v_{i,i+(k-1)n}$. Thus, the worst-case utility is maximized under truthful reporting.

Next, we prove inequality (2) (the best-case guarantee). Since agent $i$ gets exactly $\ell$ items, her utility is at most $\sum_{k=1}^{\ell} v_{i,k}$. This utility is realized when she reports $\mathbf{v}_i$, and everyone else ranks the items in the opposite order, i.e. $v_{i',j} < v_{i',j+1}$ for all $i'$, and it cannot be improved upon. $\qquad\square$

In fact, we can strengthen this result to give a "best-of-both-worlds" NOM mechanism. Specifically, we prove that the PS-Lottery algorithm of [Azi20b], which outputs randomized allocations that are ex-ante EF and ex-post EF1, is NOM in expectation; proof has been deferred to Appendix B.

## 4  Efficient Mechanisms

In this section we study whether natural efficiency notions are compatible with NOM. We consider the three most popular notions of efficiency: utilitarian social welfare, Nash social welfare and egalitarian social welfare (see Appendix C). Under truthfulness, the *only* Pareto efficient and truthful algorithm is a dictatorship, which immediately implies that one cannot truthfully achieve any non-trivial approximations with respect to any of these notions. Omitted proofs can be found in Appendix C.

**Utilitarian Social Welfare.** We start by considering the utilitarian social welfare maximizing algorithm, i.e. the algorithm which allocates each item to the agent that values it the most. When the winner for an item is not unique the algorithm needs to break ties; the choice of the tie-breaking rule will be crucial for our positive result in this section. In the context of fair division it is standard to assume that agents' values are normalized when analyzing utilitarian social welfare. Specifically, the most common assumption is that the agents' values add up to 1; see [Azi20a] for a number of justifications for this assumption. For the remainder of this section we also assume that $\sum_{j \in M} v_{i,j} = 1$ for all agents $i$.[4] We note that without this assumption, the utilitarian social welfare maximizing algorithm is NOM, since no matter what an agent reports, the best case for her is that she wins all items she values positively, and the worst case is that she loses all items, so no misreport can help increase either the worst-case or best-case utility.

**Theorem 2.** *Every (randomized or deterministic) mechanism for $n = 2$ agents that always outputs utilitarian social welfare maximizing allocations is obviously manipulable.*

Surprisingly, this impossibility result does not hold for more than two agents.

**Theorem 3.** *The utilitarian social welfare maximizing algorithm, coupled with an appropriate tie-breaking rule, is not obviously manipulable for $n \geq 3$ agents.*

*Proof.* First, we describe our algorithm. The algorithm, given reports $\mathbf{b}_1, \ldots, \mathbf{b}_n$, allocates each item to an agent with the largest reported value. In case of a tie, the item is allocated to the agent with the smallest index (i.e. $1 \succ 2 \succ 3 \ldots$), *except* if the tie is exactly between agents 1 and $n$, in which case the item goes to agent $n$. Consider an arbitrary agent $i$ with true valuation $\mathbf{v}_i$.

Towards proving inequality (1), we have that if $b_{i,j} < 1$ for all $j \in M$, there exists a choice for $\mathbf{v}_{-i}$ such that agent $i$ does not have the highest value for any item (e.g., some agent $k$ could have a value of 1 for $i$'s favorite item, and a different agent $k'$ can out-bid $i$ in all remaining items). If $b_{i,j} = 1$ for some $j \in M$, then, again, $i$ can again end up with no items, since by the choice of our tie-breaking rule, all agents can lose in the tie-breaking (agents other than 1 lose to some smaller index agent,

---

[4]Equivalently, we can define the utilitarian social welfare maximizing algorithm to be the aforementioned algorithm executed on transformed valuations where $\hat{v}_{i,j} = \frac{v_{i,j}}{\sum_{j \in M} v_{i,j}}$ for all $i \in N$.

while agent 1 loses to agent $n$). Therefore, no matter what agent $i$ reports, in the worst-case she gets a utility of zero. Next we prove inequality (2), the best-case guarantee. By the normalization assumption and tie-breaking rule, any agent $i$ (except agent 1) cannot out-bid every other agent on every single item, no matter what $\mathbf{b}_i$ and $\mathbf{v}_{-i}$ are. Therefore, the best-case outcome for agent $i$ is that she is allocated all items except her least favorite one. Additionally, note that this outcome can be realized when agent $i$ reports honestly and $\mathbf{v}_{-i}$ is such that $v_{i',j} = 1$, for all $i' \neq i$, where $j$ is the item that $i$ values the least (i.e. $j \in argmin_{k \in \mathrm{M}}\{v_{i,k}\}$). Finally, if agent 1 reports her true valuation, the best-case happens when all other agents also report agent 1's true valuation — in which case, as per the tie-breaking rule, she gets all the items; this outcome also cannot be improved upon. $\qquad\square$

**Nash Social Welfare.** The next natural maximization objective, and arguably the most popular one in the context of fair division, is the Nash social welfare. Here, we show that — unlike utilitarian social welfare maximization — there does not exist a NOM mechanism that always outputs allocations that maximize the Nash social welfare, for any number of agents. The case of $n = 2$ agents follows from Theorem 10 (Appendix E) that details our "best-of-both-worlds" results. In the following theorem (proof deferred to Appendix C), we show that the impossibility extends to $n \geq 3$ agents.

**Theorem 4.** *Every (randomized or deterministic) mechanism that always outputs an allocation that maximizes the Nash social welfare is obviously manipulable, even for $3$ additive agents and $4$ items.*

# 5 Fair and Efficient Mechanisms

Maximizing the Nash social welfare results in allocations that are fair in addition to being economically efficient [CKM+19]. However, as we established in Theorem 4, mechanisms that output MNW allocations are obviously manipulable for any number of agents.

In this section we state our main result, where we show that there is a deterministic NOM mechanism that is fair and economically efficient for additive valuations. This result is established by showing a black-box reduction from the problem of designing NOM mechanisms that output EF1 allocations to the problem of designing algorithms that output EF1 allocations. Additionally, this black-box reduction preserves Pareto efficiency guarantees. Missing proofs can be found in Appendix D.

The following theorem formally states our main result.

**Theorem 5.** *For additive valuations, there exists a black-box reduction, which preserves Pareto efficiency guarantees, from the problem of designing a NOM and EF1 mechanism, to designing an algorithm that computes clean, non-wasteful and EF1 allocations.*

By combining Theorem 5 with known algorithmic results we can get NOM mechanisms, with the same fairness and efficiency guarantees. Specifically, [GM21] give a pseudo-polynomial time algorithm that computes fractionally Pareto efficient and EF1 allocations. Fractional Pareto efficiency implies non-wastefulness, and without loss of generality we can assume that this algorithm outputs clean allocations.[5] Hence, by applying Theorem 5 we get the following application.

**Corollary 1.** *(via [GM21]). For agents with additive valuations, there exists a fractionally Pareto efficient,* EF1*, and* NOM *mechanism, that runs in pseudo-polynomial time.*

**The Reduction.** Our reduction, Mechanism 1, takes as input reported valuations $\mathbf{b} = (\mathbf{b}_1, \ldots, \mathbf{b}_n)$, and black-box access to a deterministic algorithm $\mathcal{M}^*$.[6] Our reduction requires the algorithm $\mathcal{M}^*$ to always output non-wasteful, clean EF1 allocations for every possible input valuation functions.

The reduction is based on two key ideas, first, through a careful construction of cases we ensure that if an agent $i \in \mathrm{N}$ reports valuation $\mathbf{b}_i$, then the set of allocations that can be returned by our reduction, Mechanism 1, for every possible $\mathbf{b}_{-i}$ can be precisely characterized (Lemma 3). Second, we prove a structural result (Lemma 4) concerning this set of possible output allocation. This structural result plays a central role in establishing that Mechanism 1 is not obviously manipulable. Further, we show that, by construction, our reduction always outputs fPO + EF1 allocations.

We begin by defining some notation. For each agent $i \in \mathrm{N}$, let $D_i$ be the set of goods that have strictly positive value for $i$ i.e. $D_i \coloneqq \{g \in \mathrm{M} \mid b_{i,g} > 0\}$. Let $\widehat{D}_i \coloneqq \mathrm{M} \setminus \cup_{j \neq i} D_j$ be the goods

---

[5]Starting from an fPO and EF1 allocation $A$, making each bundle $A_i$ clean preserves both properties.

[6]Algorithm $\mathcal{M}^*$ could possibly be computationally inefficient.

remaining after removing all goods desired by agents other than $i$. Let $R_i$, for each agent $i \in \mathrm{N}$, be the indicator for the event that the subsets $\{D_j\}_{j \in \mathrm{N}} \setminus \{D_i\}$ are pairwise disjoint, i.e. $R_i = 1$ iff $(\{D_j\}_{j \in \mathrm{N}}) \setminus \{D_i\}$ are pairwise disjoint, and $R_i = 0$ otherwise.

Mechanism 1 sequentially considers four cases based on the sets $\{D_j\}_{j=1}^n$ and the values $\{R_j\}_{j=1}^n$, to find a temporary allocation $A^*$. See Appendix D.2 for the pseudo-code.

**Case** I: The sets $\{D_j\}_{j=1}^n$ are pairwise disjoint (equivalently, $R_i = 1$ for all $i \in \mathrm{N}$). In this case, $A^*$ allocates the bundle $D_j$ to agent $j$ for each agent $j \in \mathrm{N}$.

**Case** II: $R_i = 1$ for exactly one agent $i \in \mathrm{N}$. This can occur if $D_i$ intersects two or more $D_j$s (and these are the only intersections among pairs of subsets in $\{D_k\}_{k=1}^n$). If allocating the bundle $\widehat{D}_i$ to agent $i$, and the bundle $D_j$ to each agent $j \in \mathrm{N}$, for each $j \neq i$ results in an EF1 allocation, then $A^*$ is set to this allocation. Otherwise, if this allocation is not EF1, $A^*$ is the allocation returned by $\mathcal{M}^*$.

**Case** III: There are exactly two agents $i, j \in \mathrm{N}$ such that $R_i = R_j = 1$. The only way this is possible is if $D_i, D_j$ intersect each other and any other pair of subsets $D_k, D_l$ where $\{k, l\} \neq \{i, j\}$, are disjoint. In this case, Mechanism 1 considers whether the set of goods $D_i \cap D_j$ are valued more by agent $i$ or agent $j$; each of these two subcases are similar to *Case* II (see Lines 12-21).

**Case** IV: None of the previous cases holds (equivalently, $R_i = 0$ for all $i \in \mathrm{N}$). In this case, $A^*$ is simply the allocation returned by $\mathcal{M}^*$.

The last step of Mechanism 1 (Line 24) is to make the bundle allocated to each agent $i$ in the temporary allocation *clean*, i.e. remove items from her bundle that she does not value. This is necessary for one of our technical lemmas (specifically, for characterizing the set of all allocations that are possible outputs of our mechanism). Note that this step does not affect efficiency or envy-freeness up to one item (i.e. if the allocation satisfied any of these notions before this step, it continues to do so).

For the case of two agents, our reduction is especially simple: give each agent $i$ all items she has a positive value and the other agent has a zero value, and give all items that both agents want (if any) to the agent with the largest value for them; if this is not EF1, run the black-box mechanism.

**The case of strictly positive valuations.** It is interesting to notice that when valuations are strictly positive, i.e. $v_{i,j} > 0$ for all $i, j$, Mechanism 1 simply outputs the allocation of $\mathcal{M}^*$ (Case IV). However, this does *not* imply that every PO and EF1 algorithm is NOM: in calculating worst-case outcomes, agents consider the possibility that others have zero valuations. This fact is critical in our proof of correctness (see the proof of Lemma 3).

**Establishing the main result.** Towards establishing Theorem 5, we begin by proving the following supporting lemmas. We show that Mechanism 1 preserves the efficiency and EF1 guarantees of $\mathcal{M}^*$.

**Lemma 1.** *If $\mathcal{M}^*$ is $\alpha$-Pareto efficient (resp. $\alpha$-fractionally Pareto efficient) then Mechanism 1 always outputs an $\alpha$-Pareto efficient (resp. $\alpha$-fractionally Pareto efficient) partial allocation.*

**Lemma 2.** *If $\mathcal{M}^*$ outputs non-wasteful, clean, and EF1 allocations then Mechanism 1 always outputs non-wasteful, clean, and EF1 partial allocations.*

The following lemma characterizes the set of allocations that Mechanism 1 could possibly return given the reported valuation of a particular agent. Specifically, we will show that every clean, non-wasteful and EF1 allocation, that is consistent with the reported valuation, is a possible output of Mechanism 1. Before stating the lemma, we define some useful notations. For any agent $i \in \mathrm{N}$ and valuation function $\mathbf{v}_i$, let $\mathrm{EF1}(i, \mathbf{v}_i)$ be the set of (partial) allocations $A = (A_1, \ldots, A_n)$ that are clean, non-wasteful and envy-free up to one item with respect to agent $i$ when her valuation function is $\mathbf{v}_i$, i.e., $\mathrm{EF1}(i, \mathbf{v}_i) = \{A \in \Pi_n(\mathrm{M}) \mid \forall g \in A_i \ v_{i,g} > 0, \text{ and } \forall j \in \mathrm{N} \text{ with } A_j \neq \emptyset, \mathbf{v}_i(A_i) \geq \mathbf{v}_i(A_j \setminus \{g\}) \text{ for some } g \in A_j, \text{ and } \mathbf{v}_i(\mathrm{M} \setminus \cup_{k \in \mathrm{N}} A_k) = 0\}$.

**Lemma 3.** *Given any agent $i \in \mathrm{N}$ and valuation function $\mathbf{v}_i$, for every allocation $A \in \mathrm{EF1}(i, \mathbf{v}_i)$ there exists a set of valuations $\mathbf{v}_{-i}$ such that Mechanism 1 on input $(\mathbf{v}_i, \mathbf{v}_{-i})$ outputs allocation $A$.*

Lemma 3 establishes that all partial allocations in $\mathrm{EF1}(i, \mathbf{v}_i)$ can possibly be returned by the mechanism if agent $i$ reports $\mathbf{v}_i$. Therefore, the problem of whether Mechanism 1 is obviously manipulable (partially) reduces to whether some set $\mathrm{EF1}(i, \mathbf{v}_i')$ has a better worst-case outcome than $\mathrm{EF1}(i, \mathbf{v}_i)$, with respect to the valuation vector $\mathbf{v}_i$. Before describing the subsequent lemma which develops this idea, we define some relevant notations. Given any set of (partial) allocations $S \subseteq \Pi_n(\mathrm{M})$, an agent $i \in \mathrm{N}$ and a valuation vector $\mathbf{v}$ define $\ell(i, \mathbf{v}, S) = \min_{A \in S} \mathbf{v}(A_i)$.

**Lemma 4.** *For any valuations $\mathbf{v}, \mathbf{v}'$ and any agent $i \in \mathrm{N}$, $\ell(i, \mathbf{v}, \mathrm{EF1}(i, \mathbf{v})) \geq \ell(i, \mathbf{v}, \mathrm{EF1}(i, \mathbf{v}'))$.*

*Proof.* Let $A^* \in \mathrm{EF1}(i, \mathbf{v})$ be an allocation such $\mathbf{v}(A_i^*) = \ell(i, \mathbf{v}, \mathrm{EF1}(i, \mathbf{v}))$, i.e. $A^*$ is a worst-case outcome for agent $i$ when her valuation is $\mathbf{v}$. We can assume, without loss of generality, that $A^*$ is a complete allocation because if we assign the unallocated items, $\mathrm{M} \setminus \cup_{k \in \mathrm{N}} A_k^*$, to agents other than agent $i$ arbitrarily then still the resultant allocation would continue to belong in $\mathrm{EF1}(i, \mathbf{v})$. This follows from the fact that $A$ is non-wasteful, $\mathbf{v}(\mathrm{M} \setminus \cup_{k \in \mathrm{N}} A_k^*) = 0$.

If $A^* \in \mathrm{EF1}(i, \mathbf{v}')$, the lemma immediately follows, since the minimum $\mathbf{v}(B_i)$, over all allocations $B \in \mathrm{EF1}(i, \mathbf{v}')$, is at most $\mathbf{v}(A_i)$. Otherwise, $A^* \notin \mathrm{EF1}(i, \mathbf{v}')$. Since $A^*$ is complete, and thereby non-wasteful, this is possible if either bundle $A_i$ is not clean wrt valuation $\mathbf{v}'$ or agent $i$ envies some other agent even after removal of one good. Denote by $A$ the allocation obtained by cleaning the bundle $A_i^*$ wrt valuation $\mathbf{v}'$, all other bundles remain as it is, i.e., $A_j = A_j^*$ for all $j \neq i$. Now if allocation $A \in \mathrm{EF1}(i, \mathbf{v}')$, the lemma follows, because $\mathbf{v}(A_i^*) \geq \mathbf{v}(A_i)$ and $\ell(i, \mathbf{v}, \mathrm{EF1}(i, \mathbf{v}'))$ is at most $\mathbf{v}(A_i)$. In the rest of the proof we will handle the case when $A \notin \mathrm{EF1}(i, \mathbf{v}')$.

Towards this we will construct an allocation $A'$ with the following properties: $(i)$ $\mathbf{v}(A_i') \leq \mathbf{v}(A_i)$, and $(ii)$ $A' \in \mathrm{EF1}(i, \mathbf{v}')$. These two properties together establish the desired inequality, since $\ell(i, \mathbf{v}, \mathrm{EF1}(i, \mathbf{v})) = \mathbf{v}(A_i^*) \geq \mathbf{v}(A_i) \geq^{(i)} \mathbf{v}(A_i') \geq^{(ii)} \ell(i, \mathbf{v}, \mathrm{EF1}(i, \mathbf{v}'))$.

Let $S^* := \{j \in \mathrm{N} \setminus \{i\} \mid A_j \neq \emptyset\}$ be the subset of agents with non-empty bundles in $A$. Define agent $j^* := \arg\max_{k \in S^*} \min_{g \in A_k} \mathbf{v}'(A_k \setminus \{g\})$, to be the agent in $S^*$ whose allocation "up to one item" has maximum value with respect to $\mathbf{v}'$, and let $g^* = \arg\min_{g \in A_{j^*}} \mathbf{v}'(A_{j^*} \setminus \{g\})$ be the favourite good with respect to $\mathbf{v}'$ that $j^*$ has. $A'$ is defined as follows. $A_i' = A_{j^*} \setminus \{g^*\}$, $A_{j^*}' = A_i \cup \{g^*\}$, and for all agents $k \in \mathrm{N} \setminus \{i, j^*\}$ we have $A_k' = A_k$. It remains to show that $A'$ satisfies $(i)$ and $(ii)$.

Towards proving $(i)$ we have $\mathbf{v}(A_i') = \mathbf{v}(A_{j^*} \setminus \{g^*\}) \leq \mathbf{v}(A_i)$, where the last inequality follows from the fact that $A \in \mathrm{EF1}(i, \mathbf{v})$. For $(ii)$, note that $\mathrm{N} \setminus \{i\}$ can be written as the union of three disjoint sets $S_1$, $S_2$ and $S_3$, where $S_1 = \mathrm{N} \setminus (S^* \cup \{i\})$, $S_2 = (S^* \setminus \{j^*\})$, and $S_3 = \{j^*\}$. We will show that in allocation $A'$ agent $i$ with valuation vector $\mathbf{v}'$, does not envy, up to one item, any of the agents in $S_1$, $S_2$, or $S_3$; this establishes that $A' \in \mathrm{EF1}(i, \mathbf{v}')$. First, for every agent $k \in S_1$ we have that $A_k' = A_k = \emptyset$, by the definition of $S^*$, so agent $i$ cannot envy any such agent. Second, given the definition of agent $j^*$, we know that for each agent $k$ in $S_2 = S^* \setminus \{j^*\}$, we have

$$\min_{g \in A_k'} \mathbf{v}'(A_k' \setminus \{g\}) = \min_{g \in A_k} \mathbf{v}'(A_k \setminus \{g\}) \qquad (A_k' = A_k \text{ for all agents except } i \text{ and } j^*)$$

$$\leq \min_{g \in A_{j^*}} \mathbf{v}'(A_{j^*} \setminus \{g\}) \qquad \text{(by the definition of } j^*)$$

$$\leq \mathbf{v}'(A_{j^*} \setminus \{g^*\}) \qquad (g^* \in A_{j^*})$$

$$= \mathbf{v}'(A_i'). \qquad (A_i' = A_{j^*} \setminus \{g^*\})$$

Therefore, agent $i$ with valuation $\mathbf{v}'$ does not envy up to one item (and, in fact, up to any item) any agent in $S_2$. Third, recall that $A \notin \mathrm{EF1}(i, \mathbf{v}')$. By definition, agent $j^*$ has the bundle that $\mathbf{v}'$ likes the most "up to one item", and $g^*$ is that most valuable (in terms of marginal value) item in $j^*$'s bundle, so $\mathbf{v}'(A_i) < \mathbf{v}'(A_{j^*} \setminus \{g^*\})$. Therefore, we have

$$\mathbf{v}'(A_i') = \mathbf{v}'(A_j \setminus \{g^*\}) \qquad (A_i' = A_j \setminus \{g^*\})$$

$$> \mathbf{v}'(A_i) \qquad (A \notin \mathrm{EF1}(i, \mathbf{v}'))$$

$$= \mathbf{v}'(A_i \cup \{g^*\} \setminus \{g^*\})$$

$$= \mathbf{v}'(A_{j^*}' \setminus \{g^*\}). \qquad (A_{j^*}' = A_i \cup \{g^*\})$$

That is, agent $i$ with valuation $\mathbf{v}'$ does not envy, up to one item, any agent in $S_3 = \{j^*\}$. Note that, allocation $A'$ is non-wasteful wrt agent $i$. Furthermore, the bundle $A_i'$ can be cleaned maintaining the fact that $A'$ is EF1 wrt agent $i$. This establishes that $A' \in \mathrm{EF1}(i, \mathbf{v}')$ and concludes the proof. $\quad\square$

Given the previous lemmas, we can show our main theorem with respect to NOM.

**Theorem 6.** *If agents' valuation functions are additive and $\mathcal{M}^*$ outputs EF1, non-wasteful and clean allocations, then Mechanism 1 is not obviously manipulable.*

*Proof.* We first show that agents cannot improve their worst-case utilities by misreporting.

Let $i \in \mathrm{N}$ be an agent whose true valuation is $\mathbf{v}_i$ and let her misreport be $\mathbf{b}_i$. By Lemma 3, for every reported valuation vector $\mathbf{b}$ of agent $i$, and for every allocation $A \in \mathrm{EF1}(i, \mathbf{b})$, there exists a valuation profile $\mathbf{v}_{-i}$ such that $A$ is the output of Mechanism 1. On the other hand, by Lemma 2, Mechanism 1 always outputs allocations $A$ that are EF1, clean, and non-wasteful. Therefore, every output allocation of Mechanism 1 is in $\mathrm{EF1}(i, \mathbf{b})$. Therefore, if agent $i$ reports $\mathbf{v}_i$ (respectively $\mathbf{b}_i$), then the set of all possible allocations of Mechanism 1 is exactly $\mathrm{EF1}(i, \mathbf{v}_i)$ (respectively $\mathrm{EF1}(i, \mathbf{b}_i)$). Overloading notation, let $A_i^*(\mathbf{v}_i, \mathbf{v}_{-i})$ be the allocation of agent $i$ in Mechanism 1, on input $(\mathbf{v}_i, \mathbf{v}_{-i})$. For the worst-case utility of agent $i$ we have $\min_{\mathbf{v}_{-i}} \mathbf{v}_i(A_i^*(\mathbf{v}_i, \mathbf{v}_{-i})) =$

$$\min_{A \in \mathrm{EF1}(i, \mathbf{v}_i)} \mathbf{v}_i(A_i) = \ell(i, \mathbf{v}_i, \mathrm{EF1}(i, \mathbf{v}_i)) \geq^{Lem. \ 4} \ell(i, \mathbf{v}_i, \mathrm{EF1}(i, \mathbf{b}_i)) = \min_{\mathbf{v}_{-i}} \mathbf{v}_i(A_i^*(\mathbf{b}_i, \mathbf{v}_{-i})).$$

This establishes Inequality (1) from the definition of NOM. To conclude the proof, it remains to show that the best-case utility of an agent is also not improved by misreporting. Note that the best case for agent $i \in \mathrm{N}$ occurs when all other agents have no value for the items, i.e. $D_j = \emptyset$ for all $j \in \mathrm{N} \setminus \{i\}$. Hence the sets $\{D_k\}_{k \in [n]}$ are pairwise-disjoint, and Mechanism 1 (via *Case* I) allocates the entirety of $D_i$ to agent $i$, which is impossible to improve upon. $\qquad\square$

The proof of the above theorem along with the supporting lemmas establishes Theorem 5. We note that that it is not possible to improve upon this result by adding ex-ante fairness guarantees; see Appendix D.3 for details.

## Acknowledgements

The authors are supported in part by an NSF CAREER award CCF-2144208, a Google Research Scholar Award, and by the Algorand Centres of Excellence program managed by Algorand Foundation. Any opinions, findings, and conclusions or recommendations expressed in this material are those of the author(s) and do not necessarily reflect the views of Algorand Foundation.

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
