## A  Missing preliminaries

**Allocations.**  A *randomized allocation* $\mathbf{R} = \{(p^z, A^z)\}_{z=1}^k$ is a probability distribution (or a lottery) over a set of integral allocations, i.e., for every $z \in [k]$, $A^z$ is an integral allocation that occurs with a probability $p^z$. The sum of the probabilities is equal to one, $\sum_{z=1}^k p^z = 1$. The integral allocations $A^1, A^2, \dots, A^k$ constitute the *support* of $\mathbf{R}$. For each randomized allocation $\mathbf{R} = \{(p^z, A^z)\}_{z=1}^k$ there exists a corresponding *expected fractional allocation* $X = \sum_{z=1}^k p^z \cdot A^z$, where for each agent $i \in \mathrm{N}$, the fraction of items given to agent $i$ is $X_i = \sum_{z=1}^k p^z \cdot A_i^z$.[7] Here $X_{i,j}$ can be thought of as the probability with which agent $i$ is allocated item $j$ in an integral allocation that is sampled from $\mathbf{R}$. For notational clarity, we will use the letters $X, Y$ to denote allocations that are fractional or integral, $A, B$ to denote allocations that are exclusively integral and $\mathbf{R}$ for randomized allocations.

**Mechanisms.**  A mechanism $\mathcal{R}$ is *randomized* if for every reported valuations $\mathbf{b}$ it outputs a randomized allocation, i.e. it returns integral allocations that are drawn from a probability distribution corresponding to a randomized allocation. Since every randomized allocation has an associated expected fractional allocation, the output of a randomized mechanism for reported valuations $\mathbf{b}$ can also be interpreted as representing a fractional allocation. We use $\mathcal{R}(\mathbf{b}) = (\mathcal{R}_1(\mathbf{b}), \dots, \mathcal{R}_n(\mathbf{b}))$ to denote the *expected fractional allocation* that a randomized mechanism outputs given bids $\mathbf{b}$; the vector $\mathcal{R}_i(\mathbf{b}) \in [0,1]^m$ represents the probabilities with which agent $i$ receives each item in the sampled integral allocation. For a randomized mechanism, $\mathbf{v}_i(\mathcal{R}_i(\mathbf{b}))$ denotes the expected utility of agent $i$ in the sampled integral allocation when the input to the mechanism $\mathcal{R}$ is $\mathbf{b}$.

To define NOM for a randomized mechanism $\mathcal{R}$, we compare the expected utilities $\mathbf{v}_i(\mathcal{R}_i(\cdot, \cdot))$ in inequality (1) and (2), instead of $\mathbf{v}_i(\mathcal{M}_i(\cdot, \cdot))$ as in the case of deterministic mechanisms. Notice that for randomized mechanisms, the definition of NOM takes an expectation over the randomness of the mechanism, and minimum/maximum are over the reports of other agents; we sometimes write "NOM in expectation" when referring specifically to a randomized mechanism.

## B  Missing from Section 3

The PS-Lottery algorithm is based on the well-known probabilistic serial algorithm, which outputs fractional allocations that are envy-free. On a high level, the PS-Lottery algorithm uses Birkhoff's algorithm[8] to implement the fractional allocation output by probabilistic serial as a randomized allocation (a lottery) over a set of EF1 allocations. For the sake of completeness, the PS-Lottery algorithm is formally described in Appendix B.1.

We begin by proving a lemma that highlights a connection between not obvious manipulability and randomized mechanisms that output ex-ante proportional allocations. We note that a similar observation is made in [OSH19] in the context of cake cutting.

**Lemma 5.** *Inequality (1) (the worst-case guarantee) is satisfied for every randomized mechanism $\mathcal{R}$ that outputs ex-ante proportional allocations.*

*Proof.* Let $\mathcal{R}$ be a randomized mechanism that outputs ex-ante proportional allocations. Consider an agent $i \in \mathrm{N}$ with true valuation $\mathbf{v}_i$.

Suppose agent $i$ reports her true valuation $\mathbf{v}_i$. Since the mechanism outputs ex-ante proportional allocations, for every possible reports of other agents, $\mathbf{b}_{-i}$, the expected fractional allocation output by $\mathcal{R}$, $Y = (Y_1, Y_2, \dots, Y_n)$ will be such that $\mathbf{v}_i(Y_i) \geq \frac{1}{n}\mathbf{v}_i(\mathrm{M})$. As a consequence, when agent $i$ reports $\mathbf{v}_i$, her worst-case expected utility is at least $\frac{1}{n}\mathbf{v}_i(\mathrm{M})$.

Next, we show that when every agent $j \in \mathrm{N} \setminus \{i\}$ reports her valuation to be $\mathbf{v}_i$ (the true valuation of agent $i$), then the worst-case expected utility of agent $i$, as per her true valuation and irrespective of her report, will be at most $\frac{1}{n}\mathbf{v}_i(\mathrm{M})$, which implies the lemma. Consider the case when every agent $j \in \mathrm{N} \setminus \{i\}$ reports her valuation to be $\mathbf{b}_j = \mathbf{v}_i$ and agent $i$ reports valuation $\mathbf{b}_i$. We know that the expected fractional allocation $Y' = (Y_1', Y_2', \dots, Y_n')$ returned by $\mathcal{R}$ will be proportional, i.e., for

---

[7] Note that multiple randomized allocations may have the same expected fractional allocation.

[8] Recall that, Birkhoff's algorithm, given a square bistochastic matrix, decomposes it into a convex combination (or a lottery) over permutation matrices.

each agent $j \in \mathrm{N} \setminus \{i\}$ we have $\mathbf{b}_j(Y_j') = \mathbf{v}_i(Y_j') \geq \frac{1}{n}\mathbf{v}_i(\mathrm{M})$. Adding up over all $j \in \mathrm{N} \setminus \{i\}$ we have that $\sum_{j \in \mathrm{N} \setminus \{i\}} \mathbf{v}_i(Y_j') = \mathbf{v}_i(\mathrm{M}) - \mathbf{v}_i(Y_i') \geq \frac{n-1}{n}\mathbf{v}_i(\mathrm{M})$. On rearranging we get the required inequality $\mathbf{v}_i(Y_i') \leq \frac{1}{n}\mathbf{v}_i(\mathrm{M})$. $\qquad\square$

Using Lemma 5 we can prove the following theorem.

**Theorem 7.** *The PS-Lottery algorithm of [Azi20b], which is ex-ante envy-free and ex-post EF1, is not obviously manipulable in expectation.*

*Proof.* Inequality (1) is implied by Lemma 5, since the PS-Lottery algorithm is ex-ante envy-free (see Appendix B.1), and therefore ex-ante proportional. It remains to prove Inequality (2).

First, it holds that the expected fractional allocation returned by the PS-Lottery algorithm, irrespective of the agents' reports, is such that each agent receives $\frac{m}{n}$ (fractional) items in total; see Property 2 in Appendix B.1. Second, the best-case for an agent $i \in \mathrm{N}$ who, without loss of generality, values items in the order $v_{i,1} \geq v_{i,2} \ldots \geq v_{i,m}$, and reports honestly, occurs when the reported valuation of other agents induce an opposite preference order on items, i.e., for each agent $j \in \mathrm{N} \setminus \{i\}$ we have $b_{j,m} \geq b_{j,m-1} \ldots \geq b_{j,1}$. In this case, agent $i$ would receive items $1, 2, \ldots, \lfloor \frac{m}{n} \rfloor$ in their entirety and a $\frac{m}{n} - \lfloor \frac{m}{n} \rfloor$ fraction of item $\lfloor \frac{m}{n} \rfloor + 1$. This allocation results in the maximum possible expected utility that agent $i$ can get subject to the constraint that she gets $\frac{m}{n}$ fraction of items, and therefore it cannot be improved upon, no matter what her report is. $\qquad\square$

## B.1 The PS-Lottery Algorithm of [Aziz, 2020b]

A square matrix $M \in [0,1]^{k \cdot k}$ is *bistochastic* iff the sum of entries in each of its rows and columns is equal to one, i.e., for each $i \in [k]$, we have $\sum_{j=1}^{k} M_{i,j} = \sum_{j=1}^{k} M_{j,i} = 1$. Additionally, a bistochastic matrix $N \in \{0,1\}^{k \cdot k}$ is a *permutation matrix* — each row and column of a permutation matrix contains exactly one entry having value one and all other entries are zero.

In essence, the PS-Lottery algorithm of [Azi20b] is based on the following two well-known algorithms:

*Birkhoff's algorithm.* Given a bistochastic matrix $M \in [0,1]^{k \cdot k}$ as input, Birkhoff's algorithm can be used to decompose, in polynomial time, the matrix $M$ into a convex combination of permutation matrices. That is, Birkhoff's algorithm outputs permutation matrices $\{M_i\}_{i=1}^{t}$ and positive real numbers $\{p_i\}_{i=1}^{t}$ such that $M = \sum_{i=1}^{k} p_i M_i$ and $\sum_i p_i = 1$; here $t = \mathcal{O}(k^2)$ is a positive integer.

*Probabilistic serial algorithm.* Given a fair division instance wherein agents have additive valuations, the probabilistic algorithm outputs a fractional allocation that is envy-free (and hence proportional). The fractional allocation output by probabilistic serial can be interpreted as the output of the following continuous procedure: starting from time $t = 0$, simultaneously, each agent start consuming items in the order of their preference (i.e., if $v_{i,j_1} \geq v_{i,j_2} \ldots, \geq v_{i,j_m}$ then the items are consumed in the order $j_1, j_2, \ldots, j_m$[9]) and at a rate of one item per unit time. If an item is fully consumed, then agents start consuming the next item as per their preference order. The algorithm terminates when all items have been consumed, which happens at time $t = \frac{m}{n}$. As a direct consequence of this, $(i)$ each agent gets exactly $\frac{m}{n}$ fraction of items at the end, and $(ii)$ the resultant fractional allocation is envy-free, since at every point agents are consuming their most-valued remaining item.

The PS-Lottery algorithm uses Birkhoff's algorithm to decompose the fractional allocation output by probabilistic serial into a convex combination over integral allocations, i.e., a randomized allocation. In addition, each integral allocation in the support of the randomized allocation is EF1.

In Section 3, towards showing that the PS-Lottery algorithm is NOM in expectation, we use the following properties of the PS-Lottery algorithm.

**Property 1.** *The randomized allocation output by the PS-Lottery algorithm is ex-ante EF and ex-post EF1.*

**Property 2.** *The total fraction of items that each agent gets in the expected fractional allocation returned by the PS-Lottery algorithm is $\frac{m}{n}$*

---

[9]ties can be broken arbitrarily

Additionally, suppose that agent $i \in N$ prefers the items in the order $v_{i,1} \geq v_{i,j_2} \ldots \geq v_{i,j_m}$, and the preference order to all the other agents is opposite, i.e., for all agents $k \in [n] \setminus \{i\}$, we have $v_{k,m} \geq v_{k,m-1} \ldots \geq v_{k,1}$, then in the fractional allocation output by probabilistic serial, agent $i$ gets the items $1, 2, \ldots, \lfloor \frac{m}{n} \rfloor$ entirely, and a fraction $\frac{m}{n} - \lfloor \frac{m}{n} \rfloor$ of the item $\lfloor \frac{m}{n} \rfloor + 1$.

## C   Missing from Section 4

*Proof of Theorem 2.*  We prove the theorem for the case of two items; the proof can be easily generalized to hold for any number of items. Let $\mathcal{M}$ be the utilitarian social welfare maximizing algorithm, coupled with any tie-breaking rule. Assume that both agents report a value of 1 for the first item, and zero for the other item, i.e. $b_{i,1} = 1$ and $b_{i,2} = 0$ for $i \in \{1, 2\}$. Given these reports, there must be an agent who gets item 1 with probability at least $\frac{1}{2}$ (if $\mathcal{M}$ is deterministic then this probability will be exactly 1); assume that this is agent 1, without loss of generality.

Now, consider the case when the true valuation of agent 1 is $\mathbf{v}_1 = (\frac{2}{3} + \epsilon, \frac{1}{3} - \epsilon)$, for some small $\epsilon > 0$. Additionally, suppose that agent 1 reports her true valuation (i.e., $\mathbf{b}_1 = \mathbf{v}_1$) and agent 2's reported value for item 1, $b_{2,1} > \frac{2}{3} + \epsilon$. In this case, the utilitarian social welfare maximizing allocation gives the first item to agent 2. Therefore, the worst-case utility of agent 1 when she reports her true valuation is at most $\frac{1}{3} - \epsilon$. Next, consider the dishonest report $\mathbf{b}_1 = (b_{1,1}, b_{1,2}) = (1, 0)$. Given this, if $b_{2,1} < 1$, then agent 1 gets the first item for a utility of $\frac{2}{3} + \epsilon$. Otherwise if $b_{2,1} = 1$, then agent 1 gets the first item with probability at least $1/2$ (this follows from our choice of agent 1), thus, her expected utility is at least $\frac{1}{2}(\frac{2}{3} + \epsilon) = \frac{1}{3} + \frac{\epsilon}{2}$. In either case, her utility is strictly larger than $\frac{1}{3} - \epsilon$, her worst-case utility under honest reporting. Therefore, $\mathcal{M}$ is obviously manipulable.  $\square$

*Proof of Theorem 4.*  We prove the theorem for the case of $n = 3$ agents and $m = 4$ items; we describe how our arguments can be adjusted to work for $n > 3$ agents and any number of items at the end of this proof. Let $\mathcal{M}$ be a (possibly randomized) mechanism that *always* outputs a Nash social welfare maximizing allocation.

Let the true valuation of agent 1 be $\mathbf{v}_1 = (3.9, 3, 2, 0.9)$. The subsequent proof has two parts: first, we will show that if agent 1 reports her true valuation, then the worst-case utility is exactly 2, and second, if agent 1 reports $\mathbf{b}_1 = (2, 2, 1, 1)$, then the worst-case would be strictly more than 2; the theorem follows.

*Worst-case utility when reporting honestly:* Consider the case where agent 1 reports her true valuation, i.e., the report $\mathbf{b}_1 = \mathbf{v}_1$. We know that mechanism $\mathcal{M}$ must output Nash social welfare maximizing allocations, and such allocations are necessarily EF1 [CKM$^+$19]; randomized $\mathcal{M}$ will output ex-post EF1 allocations. Consequently, agent 1 must be allocated at least one item, since otherwise some other agent will receive at least two items, and agent 1 will envy that agent even upon the removal of any one item. Furthermore, if agent 1 is allocated only item 4 the overall allocation can't be EF1, since agent 1 will envy (even upon the removal of any item) the agent that gets two of the first three items ($v_{1,4}$ is smaller than all other values). Thus, in the worst-case, agent 1 will get a bundle whose value is at least 2, her value for item 3. Next, we show that her worst-case utility is exactly equal to 2.

Let the reported valuations of agent 2 be $\mathbf{b}_2 = (0, 1, 0, 0)$ and of agent 3 be $\mathbf{b}_3 = (2, 0, 0, 1)$. Given this, the Nash welfare maximizing allocation is unique, and this allocation is such that agent 1 gets only item 3, for a total utility of 2. To see that this allocation is unique, first notice that item 2 must go to agent 2 (otherwise agent 2's utility, and therefore Nash social welfare, will be zero). Additionally, item 3 must go to agent 1 since she is the only agent with a non-zero value for it. The remaining items (1 and 4), which are valued only by agents 1 and 3, must be allocated in a way that the Nash social welfare of the resultant allocation is maximized. A simple case analysis shows that the unique allocation that maximizes Nash social welfare gives both items to agent 3. A consequence of having a unique Nash social welfare maximizing allocation is that mechanism $\mathcal{M}$ must output it, irrespective of whether $\mathcal{M}$ is randomized or deterministic.

*Worst-case utility when misreporting:* Consider the case where agent 1 misreports her valuation as $\mathbf{b}_1 = (2, 2, 1, 1)$. Given this report, the mechanism $\mathcal{M}$ must allocate at least one item to agent 1; otherwise its allocation would not be EF1 (ex-post EF1 for if $\mathcal{M}$ is randomized). We will show that

agent 1's allocation cannot be only item 3 nor only item 4 (i.e., a bundle of value 1 with respect to the reported values), irrespective of the reports of agents 2 and agent 3. Specifically, agent 1 must be allocated either $(i)$ at least two items, $(ii)$ only item 1, or $(iii)$ only item 2. In each of these cases, the value of the items allocated to agent 1 (as per her true valuation) will be strictly more than 2, which completes the proof.

We proceed to show that agent 1 cannot be allocated just item 3 or just item 4 in any Nash social welfare maximizing allocation. Towards a contradiction, assume that in a Nash social welfare maximizing allocation $A$, agent 1 is allocated either only item 3 or only item 4. Since agent 1 is allocated only one item, there must be some other agent who is allocated at least two items. We consider the following exhaustive cases based on the items allocated to this other agent.

*Case* I: There is an agent $j \in \{2, 3\}$ having both items 1 and 2. In this case, the allocation $A$ is not EF1 — and hence not maximizing Nash social welfare — since agent 1 envies agent $j$ (with respect to the reported valuation $\mathbf{b}_1$), even upon the removal of one item.

*Case* II: There is an agent $j \in \{2, 3\}$ who gets one item from the set of items $\{1, 2\}$ and one item from $\{3, 4\}$. Without loss of generality assume that agent $j$ is allocated items 2 and 3, and agent 1 gets item 4. We will show that in this case $A$ can never be a Nash social welfare maximizing allocation.

Since allocation $A$ maximizes Nash social welfare, it must be that transferring item 2 from agent $j$ to agent 1 does not increase the Nash social welfare, i.e., the following inequalities must hold:

$$b_{1,4} \cdot (b_{j,2} + b_{j,3}) \geq b_{j,2}(b_{1,3} + b_{1,4})$$
$$1 \cdot (b_{j,2} + b_{j,3}) \geq b_{j,2} \cdot (1 + 1) \qquad \text{(substituting } b_{1,3} = b_{1,4} = 1)$$
$$b_{j,3} \geq b_{j,2} \qquad \qquad (1)$$

Similarly, transferring item 3 from agent $j$ to agent 1 must also not increase the Nash social welfare:

$$b_{1,4} \cdot (b_{j,2} + b_{j,3}) \geq b_{j,3} \cdot (b_{1,2} + b_{1,4})$$
$$1 \cdot (b_{j,2} + b_{j,3}) \geq b_{j,3} \cdot (2 + 1) \qquad \text{(substituting } b_{1,2} = 2 \text{ and } b_{1,4} = 1)$$
$$b_{j,2} \geq 2b_{j,3} \qquad \qquad (2)$$

Combining inequalities (1) and (2) we have $b_{j,3} \geq 2b_{j,3}$, which can be true only if $b_{j,3} = 0$. However, since $A$ allocates item 3 to agent $j$, and $b_{j,3} = 0$, but $b_{1,3} > 0$, $A$ is not Nash social welfare maximizing (in fact, not even Pareto efficient).

The same argument can be extended to instances having $n > 3$ agents and $m = n + 1$ items by considering the case wherein agent 1's true valuation, $\mathbf{v}_1 = (3.9, 3, 3, \ldots, 3, 2, 0.9)$. Here, the misreported valuation which leads to an improvement in her (expected) worst-case utility is $\mathbf{b}_1 = (2, 2, 2, \ldots, 2, 1, 1)$. $\qquad \square$

## C.1 Egalitarian Social Welfare

In this section we will show that any mechanism — randomized or deterministic — that maximizes egalitarian social welfare is obviously manipulable. The *egalitarian social welfare* of an (integral or fractional) allocation $X$, denoted as $\text{ESW}(X)$, is defined as the minimum utility that any agent derives from allocation $X$, i.e., $\text{ESW}(X) = \min_{i \in \mathbb{N}} \mathbf{v}_i(x_i)$. An integral allocation is *egalitarian social welfare maximizing* iff $(i)$ it maximizes, among the set of all integral allocations, the number of agents having positive utility and $(ii)$ for any such maximal set of agents $S$, it maximizes the egalitarian social welfare, i.e., the minimum utility of agents in $S$.

Our result rules out the existence of NOM mechanisms that output leximin allocations,[10] since leximin allocations are, by definition, utilitarian social welfare maximizing.

Similar to utilitarian social welfare, when discussing egalitarian social welfare, we will assume that the valuations of agents are normalized: for every agent, the combined value for the set of all items is 1; see [AFRC$^+$16] for a thorough discussion.

---

[10] An allocation is leximin iff it maximizes the lowest utility, subject to that the second lowest utility and so on.

For the case of $n = 2$ agents, there are no NOM mechanisms that maximize egalitarian social welfare for that case; this follows from Theorem 10 in Appendix E. The following theorem establishes that this impossibility continues to hold for the case of $n \geq 3$ agents.

**Theorem 8.** *Every (randomized or deterministic) mechanism that always outputs an allocation that maximizes the egalitarian social welfare is obviously manipulable, even for $n = 3$ agents and $m = 4$ items.*

*Proof of Theorem 8.* Consider any (randomized or deterministic) mechanism $\mathcal{M}$ that maximizes the minimum utility among agents. To show that $\mathcal{M}$ is obviously manipulable, we consider an instance with $n = 3$ agents and $m = 4$ items. Let the true (normalized) valuation of agent 1 be $\mathbf{v}_1 = (0.3, 0.3, 0.3, 0.1)$.

First, we will show that if agent 1 reports honestly, then the worst-case case outcome is that she is allocated only item 4, i.e., her utility in the worst-case is $0.1$. Note that since $\mathcal{M}$ maximizes the number of agents with positive utility, and agent 1 has a positive value for all 4 items, it must allocate at least one item to agent 1; otherwise either agent 2 or 3 has more than one item, one of which could be transferred to agent 1 to make her utility positive. Now, consider the case where the reports of other agents are $\mathbf{b}_2 = (0, 0, 1, 0)$ and $\mathbf{b}_3 = (0.05, 0.05, 0.9, 0)$. Here, item 3 must be allocated to agent 2 because it is the only item that she desires. By a straightforward case analysis, the unique way in which we can allocate the remaining items (items 1, 2 and 4) to maximize the minimum utility is by allocating item 4 to agent 1 and items 1 and 2 to agent 3. Since this allocation is unique, it must be the output of $\mathcal{M}$, irrespective of whether it is deterministic or randomized. Therefore, the utility of agent 1 is $0.1$ under honest reporting, in the worst case.

Now, consider the case where agent 1 reports $\mathbf{b}_1 = (\frac{1}{3}, \frac{1}{3}, \frac{1}{3}, 0)$. Since $\mathcal{M}$ maximizes the number of agents having non-zero utility, and agent 1 positively values the first three items, she must get at least one of them: otherwise some other agent will have at least two of them, and transferring one of them to agent 1 would strictly increase the number of agents with non-zero utility. Hence, the worst-case utility of agent 1 when reporting $\mathbf{b}_1$ but her true values are $\mathbf{v}_1$ is $0.3$, her true value for each of the first three items. Since the worst-case utility improved, $\mathcal{M}$ is obviously manipulable.

The same construction can be extended to $n > 3$ agents and $m = n + 1$ items by considering the case wherein agent 1's true valuation $\mathbf{v}_1 = (\frac{1}{n} - \epsilon, \frac{1}{n} - \epsilon, \frac{1}{n} - \epsilon, \ldots, \frac{1}{n} - \epsilon, n.\epsilon)$ for any positive constant $\epsilon < \frac{1}{n(n+1)}$, and she misreports to $\mathbf{b}_1 = (\frac{1}{n}, \frac{1}{n}, \frac{1}{n}, \ldots, \frac{1}{n}, 0)$ leading to an improvement in her worst-case utility. $\square$

# D Proofs missing from Section 5

## D.1 Missing proofs

*Proof of Lemma 1.* First, note that Line 24 (making the temporary assignment clean) does not affect the utility of any agent $i$ for her own bundle. Therefore if the allocation before this step was $\alpha$-PO (respectively $\alpha$-fPO) then after Line 24 these efficiency guarantees will continue to hold. Considering all possible cases for Mechanism 1, if all agents $j \in \mathrm{N}$ receive the bundle $D_j$ (as in case I), the allocation is fractionally Pareto efficient. If all agents $j$ receive the bundle $D_j$, with the exception of a single agent $i$ that receives $\widehat{D}_i$, then all agents except $i$ have the maximum possible utility, i.e. their utility for all items $\mathrm{M}$. Furthermore, it is impossible to improve the utility of $i$ without decreasing the utility of some other agent $j$; this follows from the definition of $\widehat{D}_i$ and $D_j$. Therefore, this allocation is fractionally Pareto efficient as well. This allocation is considered in cases II and III (if the black-box algorithm $\mathcal{M}^*$ is not called), and it is the output of Mechanism 1 if it is also EF1. In the remaining cases (if the aforementioned allocation is not EF1, or we are in case IV) Mechanism 1 returns the allocation computed by the black-box algorithm $\mathcal{M}^*$ (modulo Line 24), which is an $\alpha$-PO (respectively $\alpha$-fPO) partial allocation by definition. $\square$

*Proof of Lemma 2.* Line 24 does not affect the utility of any agent for her own bundle, and since valuations are additive, removal of items could only decrease her utility for the bundle of a different agent. Therefore if the allocation before this step was EF1, it remains EF1 afterwards. This step also ensures that the allocation is clean. Next, considering all possible cases, we first have that if every agent $j \in \mathrm{N}$ receives the bundle $D_j$ (as in case I), the allocation is envy-free, and therefore EF1. In

cases II and III, Mechanism 1 either checks whether an allocation is EF1 before outputting it, or calls $\mathcal{M}^*$. In case IV $\mathcal{M}^*$ is called. Whenever $\mathcal{M}^*$ is called, then the output allocation is EF1 by definition.

Finally, note that Mechanism 1 returns non-wasteful allocations: it is easy to confirm that whenever Mechanism 1 doesn't call $\mathcal{M}^*$ its allocation is non-wasteful, while if $\mathcal{M}^*$ is called, the allocation of $\mathcal{M}^*$ the returned allocation is non-wasteful and clean, so it remains non-wasteful after Line 24.[11]  □

*Proof of Lemma 3.* Given the valuation vector $\mathbf{v}_i$ and an allocation $A = (A_1, A_2, \ldots, A_n) \in \text{EF1}(i, \mathbf{v}_i)$ we will construct valuations for the other agents, $\mathbf{v}_{-i}$, such that Mechanism 1 outputs $A$ on input $(\mathbf{v}_i, \mathbf{v}_{-i})$. The valuation of each agent $j \in (\text{N} \setminus \{i\})$ will be such that $v_{j,g} = 0$ for all $g \notin A_j$, and $v_{j,g} > 0$ for each good $g \in A_j$ (the precise value of $v_{j,g}$ will depend on how $D_i$ intersects with $A_j$). That is, for each agent $j \in (\text{N} \setminus \{i\})$, $D_j = A_j$. Therefore, by construction, the subsets $\{D_j\}_{j=1}^n \setminus \{D_i\}$ are pairwise disjoint, and therefore $R_i = 1$ (so, we are never in case IV of Mechanism 1). Also, by construction, Line 24 will not affect any bundle. In the rest of the proof, we consider the three exhaustive cases, based on how many sets from $A_{-i} = (A_1, \ldots, A_{i-1}, A_{i+1}, \ldots, A_n)$ the set $D_i$ intersects.

The first case is when $D_i$ does not intersect any bundle from $A_{-i}$. Since $v_{i,g} = 0$ for every unallocated good $g$ (by the definition of $\text{EF1}(i, \mathbf{v}_i)$), $D_i \subseteq A_i$, and therefore $R_j = 1$ for all $j \in \text{N}$. In this case we can set $v_{j,g}$ to be an arbitrary value for all $g \in A_j$, e.g. $v_{j,g} = 1$, for all agents $j \neq i$. Therefore, Mechanism 1 considers *Case* I, and it allocates the set $D_j$ to every agent $j$. By construction, for all $j \neq i$, $D_j = A_j$. Additionally, for agent $i$, it must be that the bundle $A_i = D_i$, since $A_i$ is also clean (by Line 24). Therefore, the output of Mechanism 1 is precisely $A$.

The second case is that $D_i$ intersects exactly one bundle from $A_{-i}$. Let $j^* \in (\text{N} \setminus \{i\})$ be the (unique) agent such that $A_{j^*} \cap D_i \neq \emptyset$. Additionally, assume that $i < j^*$; an almost identical argument works if $i > j^*$. For each agent $k \in \text{N} \setminus \{i, j^*\}$, we define $v_{k,g} = 1$ if good $g \in A_k$ and $v_{k,g} = 0$ otherwise. The valuations of agent $j^*$ are as follows

$$v_{j^*,g} = \begin{cases} 2\mathbf{v}_i(D_i \cap D_{j^*}) & g \in A_{j^*} \cap D_i \\ 1 & g \in A_{j^*} \setminus D_i \\ 0 & otherwise \end{cases}$$

Note that, the set of desired goods $D_k = A_k$ for all agents $k \in (\text{N} \setminus \{i\})$, and $\widehat{D}_i \supseteq A_i$. Furthermore, since $A$ is non-wasteful (by the definition of $\text{EF1}(i, \mathbf{v}_i)$), goods in $\widehat{D}_i \setminus A_i$ must have zero value for $i$. Therefore, $\mathbf{v}_i(\widehat{D}_i) = \mathbf{v}_i(A_i)$. The construction satisfies that the bundles $\{D_k\}_{k \in \text{N} \setminus \{i\}}$ are pairwise disjoint, and the bundle $D_i$ only intersects $D_{j^*} = A_{j^*}$. Therefore, $R_i = R_{j^*} = 1$ and $R_k = 0$ for every other agent $k$. Therefore, Mechanism 1, given valuations $(\mathbf{v}_i, \mathbf{v}_{-i})$, considers *Case* III (Lines 11-21) to compute the final allocation. In *Case* III, the mechanism first checks whether $\mathbf{v}_i(D_i \cap D_{j^*}) < \mathbf{v}_j(D_i \cap D_{j^*})$ holds, which is true for our construction since $v_{j^*,g} = 2\mathbf{v}_i(D_i \cap D_{j^*})$ for all $g$ in $A_{j^*} \cap D_i = D_{j^*} \cap D_i$. Afterwards, the mechanism checks if the allocation $(D_1, \ldots, \widehat{D}_i, \ldots, D_{j^*}, \ldots, D_n)$ is EF1 (Line 13). Since $\mathbf{v}_i(\widehat{D}_i) = \mathbf{v}_i(A_i)$, and $A \in \text{EF1}(i, \mathbf{v}_i)$, this allocation is indeed EF1 for agent $i$, and since all other agents are envy-free, Mechanism 1 sets it as the temporary assignment. After removing all zero valued items from $\widehat{D}_i$ (Line 24) we are left with the bundle $A_i$, hence, the final allocation is exactly $A$.

The third and final case is that $D_i$ intersects more than one bundle from $A_{-i}$. Here, the valuation vector of every agent $j \in \text{N} \setminus \{i\}$ is such that $v_{j,g} = 1$ for $g \in A_j$ and $v_{j,g} = 0$ otherwise. By construction, the set of desired goods $D_j = A_j$ for every agent $j \in \text{N} \setminus \{i\}$, $\widehat{D}_i \supseteq A_i$, and since $A$ is non-wasteful, we have $\mathbf{v}_i(\widehat{D}_i) = \mathbf{v}_i(A_i)$. $D_i$ intersecting more than one bundle from $A_{-i}$, is equivalent to $D_i$ intersecting more than one subsets from $\{D_j\}_{j \neq i}$, and therefore $R_j = 0$ for all $j \in \text{N} \setminus \{i\}$. Thus, given the valuation profile $(\mathbf{v}_i, \mathbf{v}_{-i})$, Mechanism 1 considers *Case* II (Lines 6-10), where it first checks whether the allocation $(D_1, D_2, \ldots, \widehat{D}_i, \ldots, D_n)$ is EF1. This allocation has the same value as $A$ for agent $i$ (and all other agents have exactly the same allocation), so agent $i$ has an envy of at most one item by definition, and all other agents have no envy at all. Therefore,

---

[11]Note that if the allocation of $\mathcal{M}^*$ was non-wasteful and not clean, then Line 24 could have altered this allocation into a wasteful one.

Mechanism 1 sets this as the temporary assignment. After removing all zero valued items from $\widehat{D}_i$ (Line 24) the final allocation is exactly $A$. This concludes the proof of Lemma 3. $\qquad\square$

## D.2 Pseudo-code for Mechanism 1

---

**REDUCTION 1:** Black-box reduction

  **Input:** Reported valuation functions of agents $\mathbf{b}$. Black-box access to an algorithm $\mathcal{M}^*$.
  **Output:** A partial integral allocation $A = (A_1, \ldots, A_n)$
  1: **Set** $D_i \leftarrow \{g \in \mathrm{M} \mid b_{i,g} > 0\}$.
  2: **Set** $\widehat{D}_i \leftarrow \mathrm{M} \setminus \cup_{j \neq i} D_j$ for each $i \in \mathrm{N}$.
  3: **Set** $R_i \leftarrow 1$ if subsets $\{D_j\}_{j=1}^n \setminus \{D_i\}$ are pairwise disjoint; 0 otherwise.
  4: **if** the subsets $\{D_j\}_{j=1}^n$ are pairwise disjoint **then**                                    $\}$ *Case* I
  5:     $A^* \leftarrow (D_1, \ldots, D_n)$
  6: **else if** $\exists i \in \mathrm{N}$ such that $R_i = 1$ and $R_j = 0$ for every $j \in \mathrm{N} \setminus \{i\}$ **then**
  7:     **if** the allocation $(D_1, D_2, \ldots, \widehat{D}_i, \ldots, D_n)$ is EF1 **then**            $\}$ *Case* II
  8:         $A^* \leftarrow (D_1, D_2, \ldots, \widehat{D}_i, \ldots, D_n)$
  9:     **else**
 10:         $A^* \leftarrow \mathcal{M}^*(\mathbf{b}_1, \mathbf{b}_2, \ldots, \mathbf{b}_n)$
 11: **else if** $\exists i, j \in \mathrm{N}$ such that $i < j$ and $R_i = R_j = 1$, and $R_k = 0$ for $k \in [n] \setminus \{i, j\}$ **then**
 12:     **if** $\mathbf{b}_i(D_i \cap D_j) < \mathbf{b}_j(D_i \cap D_j)$ **then**
 13:         **if** $(D_1, \ldots, \widehat{D}_i, \ldots, D_j, \ldots, D_n)$ is EF1 **then**
 14:             $A^* \leftarrow (D_1, \ldots, \widehat{D}_i, \ldots, D_j, \ldots, D_n)$
 15:         **else**
 16:             $A^* \leftarrow \mathcal{M}^*(\mathbf{b}_1, \mathbf{b}_2, \ldots, \mathbf{b}_n)$            $\}$ *Case* III
 17:     **else**
 18:         **if** $(D_1, \ldots, D_i, \ldots, \widehat{D}_j, \ldots, D_n)$ is EF1 **then**
 19:             $A^* \leftarrow (D_1, \ldots, D_i, \ldots, \widehat{D}_j, \ldots, D_n)$
 20:         **else**
 21:             $A^* \leftarrow \mathcal{M}^*(\mathbf{b}_1, \mathbf{b}_2, \ldots, \mathbf{b}_n)$
 22: **else**                                                                                               $\}$ *Case* IV
 23:     $A^* \leftarrow \mathcal{M}^*(\mathbf{b}_1, \mathbf{b}_2, \ldots, \mathbf{b}_n)$
 24: For all $i \in \mathrm{N}$, iteratively remove goods $g \in A_i^*$ such that $\mathbf{b}_i(A_i^* \setminus \{g\}) = \mathbf{b}_i(A_i^*)$.
 25: **return** $A^*$

---

## D.3 Ex-ante EF, Ex-post fPO, Ex-post EF1 and NOM

In the previous section we established (Application 1) the existence of a deterministic NOM mechanism that outputs ex-post fPO and EF1 allocations for additive agents. Here we make a short remark that it is not possible to improve upon this result by adding ex-ante fairness guarantees. Specifically, there does not exist a randomized mechanism that is NOM in expectation and outputs ex-post fPO and EF1 allocations, that is additionally ex-ante envy-free. This follows directly from the following impossibility result of [FSV20].

**Theorem 9** ([FSV20])**.** *There exists instances with additive valuation where there is no randomized allocation that is simultaneously ex-post fPO and* EF1*, and ex-ante envy-free.*

## E  Best of Both Worlds

In the previous sections we saw how fairness guarantees can be turned into NOM guarantees. For example, Lemma 5 says that the worst-case guarantee of NOM is satisfied for ex-ante proportional algorithms, while Theorem 5 turns any EF1 algorithm into a NOM (plus EF1) mechanism. In this section we show that this connection can be exploited in the other direction to prove impossibility results for fair algorithms. Specifically, we show that certain "best-of-both-worlds" fairness guarantees,

that is, randomized allocations that satisfy an ex-ante and ex-post guarantees simultaneously, are not guaranteed to exist. Our impossibility result uses the following theorem.

**Theorem 10.** *Every deterministic (or randomized) mechanism for $n = 2$ agents with additive and normalized utilities, that always outputs allocations that (ex-post) maximize the number of agents having positive utility, is obviously manipulable.*

*Proof of Theorem 10.* Let $\mathcal{M}$ be such a deterministic (or randomized) mechanism. We will prove the theorem (i.e., show that $\mathcal{M}$ is obviously manipulable) for the case of $m = 2$ items; the proof easily generalizes to more than 2 items.

Consider the case when both the agents report a value of $1$ for the first item and zero for the other item, i.e. $\mathbf{b}_i = (1, 0)$ for $i \in \{1, 2\}$. Given these reports, there must be an agent who gets item 1 with probability at least $\frac{1}{2}$ (if $\mathcal{M}$ is deterministic then this probability will be exactly 1); assume that this is agent 1, without loss of generality.

Consider the case that the true valuation of agent 1 is $\mathbf{v}_1 = (\frac{2}{3} + \epsilon, \frac{1}{3} - \epsilon)$, for some small constant $\epsilon > 0$. The worst case for agent 1 under honest reporting happens when the reported valuation of agent 2 $\mathbf{b}_2 = (1, 0)$, in which case item 1 must be allocated to agent 2 and item 2 to agent 1 – irrespective of whether the mechanism $\mathcal{M}$ is deterministic or randomized since $\mathcal{M}$ maximizes the number of agents having positive utility *ex-post*, and allocating item 1 to agent 2 and item 2 to agent 1 is the unique allocation that gives both agents non-zero utility. Therefore, the worst-case utility of agent 1 is at most $\frac{1}{3} - \epsilon$.

Now, consider the dishonest report $\mathbf{b}_1 = (b_{1,1}, b_{1,2}) = (1, 0)$. If $b_{2,2} > 0$, then mechanism $\mathcal{M}$, in order to maximize the number of agents having positive utility, will allocate item 2 to agent 2 and item 1 to agent 1, i.e., agent 1's utility in this case (per $\mathbf{v}_1$) would be $\frac{2}{3} + \epsilon$. On the other hand, if $b_{2,2} = 0$ (and therefore $b_{2,1} = 1$) then as per our choice of mechanism $\mathcal{M}$, agent 1 will receive item 1 with probability at least $\frac{1}{2}$. Hence, her utility in this case would be at least $\frac{1}{2}\mathbf{v}_{1,1} = \frac{1}{3} + \frac{\epsilon}{2}$. The above case analysis shows that if agent 1 reports valuation $\mathbf{b}_1$ then her worst-case utility will be $\frac{1}{3} + \frac{\epsilon}{2}$, which is strictly larger then her worst-case utility when reporting true valuation. Therefore, $\mathcal{M}$ is obviously manipulable. $\square$

We note that we can apply Theorem 10 to show that for $n = 2$ there is no NOM mechanism that outputs (ex-post) and MNW solution, since MNW is scale free (and therefore normalization comes without loss of generality).

We are now ready to prove our main result for this section.

**Theorem 11.** *Under additive valuations, randomized allocations that are ex-ante proportional, ex-post Pareto efficient and ex-post maximize the number of agents having positive utility do not exist, for any number of agents and any number of items.*

*Proof.* We begin by establishing the result for $n = 2$ agents. Towards a contradiction, assume that for additive valuations, randomized allocations that are ex-ante proportional, ex-post Pareto efficient and ex-post maximize the number of agents having positive utility always exits, and let $\mathcal{R}$ be a randomized mechanism that, given the reported valuations of agents, outputs such a allocation. Note that this mechanism need not be computationally efficient. We will show that $\mathcal{R}$ is NOM in expectation, in direct contradiction to Theorem 10.

Since, $\mathcal{R}$ outputs ex-ante proportional allocations, the worst-case guarantee of NOM (inequality (1)) holds, via Lemma 5. Additionally, the best-case guarantee (inequality (2)) is also satisfied: if an agent $i \in M$ reports her true valuation, then in the best case, all other agents only desire her least-valued item; in this case, agent $i$ receives all items except her least-valued item. This outcome cannot be (strictly) improved as agent $i$ must lose at least one item: the mechanism maximizes the number of agents having positive utility and each agent reports having a positive utility of at least one item, therefore, all items cannot be allocated to the same agent. Therefore, $\mathcal{R}$ is NOM is expectation, a contradiction.

Thus, for $n = 2$ agents randomized allocation that are ex-ante proportional, ex-post Pareto efficient, and ex-post maximize the number of agents having positive utility do not always exist. In particular, there is a 2 agent instance where such randomized allocations do not exist. This instance can be easily converted into a $n > 2$ agent instance by adding dummy agents $a_1, a_2, \ldots, a_k$ and dummy

items $j_1, j_2, \ldots, j_k$ such that $(i)$ for each $i \in [k]$, agent $a_i$'s value for item $j_1$ is 1 and her value for every other item is 0, $(ii)$ the existing agents have zero value for each dummy item. In this new instance, every Pareto efficient allocation must be such that agent $a_i$ is allocated item $j_i$ and no other item that existing agents value positively. Thus, a randomized allocation that is ex-ante proportional, ex-post Pareto efficient, and ex-post maximizes the number of agents having positive utility would continue to satisfy these properties after removing the dummy agents and dummy items. However, as already established, such allocations do not exist for $n = 2$ agents. This concludes the proof of Theorem 11. □

The existence of ex-ante proportional, ex-post Pareto efficient and ex-post EF1 allocations remains an elusive open problem. Our result shows that replacing ex-post EF1 with a different, mild fairness guarantee, namely ex-post maximizing the number of agents with positive utility, is impossible. The following corollary is immediate (and we note that point #3 in the following corollary is shown by [FSV20]).

**Corollary 2.** *Under additive valuations, randomized allocations that satisfy the following properties do not always exist.*

1. *ex-ante proportional, ex-post Pareto efficient and ex-post egalitarian welfare maximizing*

2. *ex-ante proportional and ex-post leximin allocations*

3. *ex-ante proportional and ex-post MNW.*