# OpenReview forum: "Fair and Efficient Allocations Without Obvious Manipulations"
_NeurIPS.cc/2022/Conference — NeurIPS 2022 Accept_

### Official Review · Reviewer_FBTg · 2022-07-11

**Rating:** 4
**Confidence:** 4
**Soundness:** 3 good
**Presentation:** 4 excellent
**Contribution:** 2 fair

**Summary:**

This paper reconsiders the classic problem of allocating indivisible goods through the lens of non-obvious manipulability (NOM).  The results show that this can bypass known impossibility results for the stronger and more standard requirement of truthfulness.  In particular is possible to achieve deterministic+EF1+NOM and that this can be combined with (fractional) Pareto efficiency among other positive results.  Negative results show that Nash and egalitarian welfare maximization both fail to satisfy NOM.

**Questions:**

Can you make a case that the stronger theoretical properties yielded by the reduction lead to meaningfully better results in practice.

**Limitations:**

NOM is a relatively recent approach.  Given that the interest of the results hinges on the reasonableness of relaxing truthfulness to NOM, I was surprised there does not appear to be discussion of whether the guarantee provided by NOM and the incentives of them mechanisms that satisfy it are reasonable in this setting.

**Strengths And Weaknesses:**

On the positive side, the paper is clear and does a nice job revisiting a classic problem and gaining new insights about both classic mechanisms and new designs.  The black-box-reduction seems technically non-trivial and from a theoretical perspective the combination of properties it achieves is an improvement over prior work.

On the negative side, the arguments in Sections 3 and 4 largely seem straightforward, so the contribution here seems somewhat limited.  For Section 5, which has more of a technical contribution, it is unclear to me how important this is for practice.  Other than identifying that it has stronger theoretical properties, there isn’t any evidence presented that the results of the reduction are noticeably better than Round-Robin or PS-Lottery.

---

> ### Author Response · Authors · 2022-07-31
> **Response to Reviewer FBTg**
>
> Thank you for the thoughtful review and question. Please see our response to all reviewers for your comment about the weakness of NOM as a definition.
>
> - "Can you make a case that the stronger theoretical properties yielded by the reduction lead to meaningfully better results in practice."
>
> See our response to all reviewers for more justification on why NOM is interesting/meaningful.
> One way our results, and specifically our reduction can inform practice is the following, concrete message. By adding a few lines of code (implementing the first 3 steps of the reduction; Cases I, II and III in lines 320-329) to the implementation of any EF1 + PO algorithm (e.g. MNW, which is used by the popular website Spliddit) one can *provably* protect against certain deviations (or, equivalently, strategic but not perfectly rational and all-knowing agents).

---

> > ### Comment · Reviewer_FBTg · 2022-08-04
> > **Response to Response**
> >
> > While incorporating aspects of the overall and specific response into the paper would be beneficial, I still feel that neither addresses the core substance of the relevant concerns.  On the reasonableness of NOM, the case presented is generic.  But nothing in the response or the paper seems to address whether they are reasonable in this specific setting.  Do these mechanisms that satisfy NOM have manipulations that, while not technically satisfying the definition of obvious, might be called such by a lay reader?  The response details how Troyan and Morrill argue this in a different setting, and not at least attempting to make such a case here seems a real limitation.
> >
> > Regarding the reduction, suppose I do not care about NOM.  Are there any other reasons to prefer it to Round Robin?  Or tying this back to the former can you identify a natural class of examples where NOM provides better results than a manipulated round robin?  The mechanism seems substantially more complex to interact with and explain to users (both because of the more complex underlying algorithm and that added by the reduction) than round robin, so I'm really looking for this sort of justification for whether it has benefits in practice.

---

> > > ### Author Response · Authors · 2022-08-05
> > > **Response**
> > >
> > > Thank you for your comment. Some of our general comments apply to the specific setting studied here, but let us clarify more, as well as expand on some of these points.
> > >
> > > - “While incorporating...a real limitation.”
> > >
> > > First, regarding your specific question about whether mechanisms that satisfy NOM have manipulations that might seem obvious, but don’t fit the definition, the answer is quite subjective and context specific. In our setting, when participating in Round Robin (RR), if I *know* that my favorite item is ranked last by everyone else, it seems obvious that I shouldn’t select it first, but I should instead pick my second favorite item. This is not an obvious manipulation, according to the definition of obvious, since it requires detailed knowledge of others’ preferences. We do not have data that confirms or refutes whether this occurs in practice, and without a formal model about what is and what is not obvious/reasonable it’s hard to provide anything but a subjective opinion. Our subjective opinion is that, when agents know a bit about each other, and when such a situation occurs (my favorite is ranked low for others), this manipulation is very reasonable, and the designer should worry about such things. Our subjective opinion is also that when participating in a second-price auction everyone should report the truth, but the practical evidence says otherwise (see [Li17]).
> > >
> > > Connecting the previous point to the reasonableness of NOM in our setting (in theory and practice), lack of knowledge about others’ preferences in practice, the fact that makes the above deviation in RR non-obvious, and one of the core reasons of why studying NOM in our setting is worthwhile (see our original response), is quite literally one of the two justifications given in [Caragiannis et al. 2019] for why manipulations (in MNW) are not a major concern in Spliddit, a popular platform for allocating indivisible goods (i.e. the same setting as here). This also serves as an example of why we need these formal models: MNW *can* be manipulated in obvious ways (according to the given definition of obvious) contradicting the intuition/informal argument of Caragiannis et al.
> > >
> > > Interestingly, the other justification for why Caragiannis et al. do not address incentives is that truthfulness rules out reasonable algorithms in this setting (that is, implicitly the authors are saying “since we can’t get truthfulness, we have to settle for no guarantees on manipulations”); the lack of more nuanced guarantees between “no manipulation is possible, ever” (i.e. truthfulness) and “any manipulation could be an issue” (non-truthfulness) is another core reason of why studying NOM in our setting is worthwhile.
> > >
> > > - “Regarding...in practice.”
> > >
> > > One might want to avoid using RR in practical settings because, even though it is simple, it is very inefficient: it does not even guarantee a constant approximation to *Pareto* efficiency (so, let alone concrete objectives, like sum/product of utilities): consider the case where agent 1 wants all items equally at a value of 1 (except the first item with a value of 1+epsilon) and agent 2 wants the first item for a large value and all the other items at epsilon. RR would give the same number of items to each agent, with the first item going to agent 1. Agent 2 would be happy to trade all her items for the first item, vastly improving her utility, and also doubling the utility of agent 1. A similar example for more agents gives a super constant improvement to everyone.
> > >
> > > A closely related and very reasonable question is why should we not use MNW, which is EF1 + PO but not NOM, if we don’t care about NOM? (Note that MNW is NP-hard to compute, but it is arguably a simpler EF1 + PO algorithm to explain compared to polytime algorithms with the same guarantees) Indeed, if one is truly not worried about *any* manipulation, there is no reason to use our reduction. To reuse the argument of Caragiannis et al., one might not worry about manipulations in MNW because agents don’t know each others’ preferences. However, as we show in this paper, this is not correct: even if participants know nothing about each other, there are (formally speaking obvious) deviations. Roughly speaking, if there are n agents and n items, if I only like two items, I should *obviously* misreport and declare a positive value for only my favorite: MNW wants to give positive utility to me, to avoid getting an objective of zero, so I’m forcing it to give me my favorite item. Our reduction says that adding three lines of code provably protects against such deviations. We don’t see this as adding complexity or majorly impactful in terms of explainability, since the current version of MNW employed in practice also doesn’t simply maximize the product of utilities, but first handles corner cases (namely, if the optimal product is zero, it first maximizes the number of agents that get positive utility, followed by MNW on the chosen agents).

---

> > > > ### Comment · Reviewer_FBTg · 2022-08-07
> > > > **Response**
> > > >
> > > > Thanks for the additional discussion of these issues.  I think bringing discussion along these lines into appropriate places in the paper will strengthen it.

---

> > > > > ### Author Response · Authors · 2022-08-08
> > > > > **Response**
> > > > >
> > > > > We will certainly incorporate this discussion in the final version. We hope you will reconsider your score in light of the response. Please let us know if you have any further questions.

---

### Official Review · Reviewer_uv2Y · 2022-07-11

**Rating:** 8
**Confidence:** 3
**Soundness:** 4 excellent
**Presentation:** 4 excellent
**Contribution:** 3 good

**Summary:**

This work considers the tools available to a central designer needing to allocate goods fairly and efficiently to agents with additive valuation functions and no monetary transfers. There are no general envy-free and deterministic truthful mechanisms to do so. Recent other work in loosening truthfulness has explored what types of manipulations are likely to be taken advantage of. This work takes one such notion, “non-obvious manipulability” and asks what additional strength does relaxing truthfulness to NOM give to the central designer - notably, can she achieve a mechanism that  is deterministic and envy-free up to one item? More generally, how do her options change when she relaxes truthfulness to NOM?

The paper answers in the affirmative, and in fact gives a black box reduction from designing an EF1+NOM mechanism to that of designing an EF1 algorithm.

This black box reduction hinges on handling the special cases where the demanded items are fully disjoint sets (1), fully disjoint except for the goods desired by one person (case 2), fully disjoint except the goods desired by two people (case 3), and if none of those apply, applying the algorithm directly (case 4).




**Questions:**

Can you comment on the jump from 2 to 3 agents in both Theorems 2&3 and in the core reduction and result? Does NOS retain relevance with three agents having overlapping preferences or is this a case where it is capturing

**Strengths And Weaknesses:**

The paper comprehensively addresses the impact that relaxing truthfulness to non-obviously manipulability gives to a designer. This makes for a helpful and interesting contribution to the literature surrounding relaxed notions of truthfulness.

The nature of the black box reduction - special cases for when two or fewer agents have overlapping demand sets raises a small question of whether or not there is a more fundamental core regarding three agents that can be embedded to simplify the reduction.

---

> ### Author Response · Authors · 2022-07-31
> **Response to Reviewer uv2Y**
>
> Thank you for the thoughtful review and question.
>
> - “Can you comment on the jump from 2 to 3 agents in both Theorems 2&3 and in the core reduction and result? Does NOS retain relevance with three agents having overlapping preferences or is this a case where it is capturing”
>
> Let us clarify. Regarding the jump from 2 to 3 in Theorems 2 & 3 (utilitarian welfare), the issue is tie-breaking. With 2 agents, a mechanism will end up tie-breaking in favor of one of the two, giving this winner an obvious deviation in certain scenarios. Fortunately, this corner case happens to be the only obstacle to achieving NOM. And, with 3 or more agents, this can be avoided by setting up a cyclical tie-breaking rule (1 loses to 2, 2 loses to 3, … n loses to 1) that avoids the unique, consistent winner (and therefore, for all agents, in the worst-case the deviation won’t work because they’ll be faced with an agent that beats them).
> The reduction works for any number of agents (also see our response to reviewer iRAr for an alternative presentation of the reduction). In the case of 2 agents, it would have an especially simple form, where we simply check if allocating the items in the intersection (if any) to the agent with the largest value for these items is EF1, and if not run the black-box. Thank you for this interesting observation.

---

### Official Review · Reviewer_iRAr · 2022-07-11

**Rating:** 7
**Confidence:** 3
**Soundness:** 4 excellent
**Presentation:** 3 good
**Contribution:** 3 good

**Summary:**

This paper studies the problem of non-monetary resource allocation when agents have additive valuation functions over indivisible goods. Since there exist strong impossibility results in this setting if one also imposes truthfulness, the authors consider the relaxed incentive guarantee known as non-obvious manipulability (NOM). The main result is that any algorithm that satisfies envy-freeness up to one good (EF1) can be transformed into a not-obviously manipulable and EF1 algorithm (subject to mild conditions). This main result is complemented by some other results regarding NOM in the domain of non-monetary resource allocation: the round robin and utilitarian social welfare-maximizing algorithms are NOM, but Nash and egalitarian social welfare-maximizing algorithms are not.

**Questions:**

1) Lines 249-250 grabbed my attention, since my intuition is that manipulating the utilitarian SW-maximizing mechanism is (intuitively) "obvious" when valuations are not required to be normalized: I should report huge valuations for all goods. Is there intuition for why we shouldn't think of this as an "obvious" manipulation? It would also help me if you could lay out the formal relationship between NOM and obvious strategyproofness of Li17, if there exists one. Lines 38-40 hint at such a relationship but leave it ambiguous.

2) Can you write down the mechanism that is obtained from applying the reduction (Mechanism 1) to MNW? Even just for 3 agents and 4 items?

**Limitations:**

Adequately addressed.

**Strengths And Weaknesses:**

Strengths: The idea of the paper is novel and the idea is very interesting. Escaping and refining classic impossibilities in this space is an important problem, in my opinion. The results that are obtained, especially the main result (Theorem 5), are strong and surprising.

Weaknesses: I am not convinced that NOM is a good relaxation of truthfulness (for this problem, at least). See my question below.

Minor:

Line 99: "efficiency" -> "efficient"
Line 184: "an" -> "a"
Line 191: "agent" -> "agents"
Could lines 197-202 go to the appendix?

---

> ### Author Response · Authors · 2022-07-31
> **Response to Reviewer iRAr**
>
> Thank you for the thoughtful review and questions. Please see our response to all reviewers for your comment about the weakness of NOM as a definition.
>
> - “Lines 249-250 grabbed my attention, since my intuition is that manipulating the utilitarian SW-maximizing mechanism is (intuitively) "obvious" when valuations are not required to be normalized: I should report huge valuations for all goods. Is there intuition for why we shouldn't think of this as an "obvious" manipulation?”
>
> Indeed, it would seem that utilitarian welfare maximization is obviously manipulable since overreporting definitely dominates telling the truth. However, in the worst case, no matter how much you overreport your values, it is not enough, and you won’t win a single item. And, in the best case, reporting the truth (or anything positive really) is enough to get you all items you want. This phenomenon is perhaps akin to a first price auction, where bidding your value is definitely a bad idea, but it is not clear how much lower than your value you should bid (and in the worst case, for all $\epsilon$, underbidding by $\epsilon$ was too low).
>
> - “It would also help me if you could lay out the formal relationship between NOM and obvious strategyproofness of Li17, if there exists one. Lines 38-40 hint at such a relationship but leave it ambiguous.”
>
> [Li17] defines what it means for a strategy to be obvious. He then uses this definition to define what it means for a mechanism to be obviously truthful (if it has an equilibrium in obviously dominant strategies), a requirement more strict than truthfulness (equilibrium in dominant strategies). [Li17] points out that even though many mechanisms are truthful, in practice it is not always easy for real people to figure out their dominant strategy. One of the interesting properties of Li’s definition is that it manages to formally separate truthful mechanisms in a way that is consistent with practical evidence, e.g. an ascending auction and a second price auction (empirically, it is much easier for real people to figure out how to play in an ascending auction vs a second price auction).
> Here, we use exactly the same definition of “obvious” for a strategy. And, similarly to [Li17], our paper (following [Troyan and Morrill, 2020]) aims to separate non-truthful mechanisms in terms of how easy it is to find a profitable deviation. We aim to identify and design mechanisms that might be non-truthful, but finding a profitable deviation is not obvious.
>
> - "Can you write down the mechanism that is obtained from applying the reduction (Mechanism 1) to MNW? Even just for 3 agents and 4 items?"
>
> A perhaps simpler way to look at the reduction is the following:
> 1) First, check if all agents want distinct items. If so, we are done.
> 2) For all $i \in [n]$, check if removing $i$ makes all remaining agents want distinct items. If $i$ is the unique such agent, temporarily assign everyone else the items they want. If $i$ is happy (in the EF1 sense) to be allocated all unclaimed items, we are done.
> 3) If there are exactly two agents, $i$ and $j$, whose desired sets overlap (but everyone else wants distinct stuff), temporarily assign everyone else the items they want. Give $i$ or $j$ the items they want, depending on who wants the items in the intersection more. Give the remaining agent everything that’s left. If this is EF1 we are done.
> 4) If all previous steps failed, run the black box algorithm (e.g., MNW).
>
> Overall, we believe it is easier to think of the reduction as taking care of some extreme cases in an engineered way (to guarantee NOM), followed by calling the black box if the input is not in the extreme cases. In our view, this simplicity (in terms of the coding overhead) is a feature: we can very easily take an implementation of an EF1 + PO algorithm and turn it into an EF1 + PO + NOM mechanism.

---

### Official Review · Reviewer_Hf6D · 2022-07-13

**Rating:** 6
**Confidence:** 3
**Soundness:** 3 good
**Presentation:** 2 fair
**Contribution:** 3 good

**Summary:**

Truthfulness is an important issue in the field of indivisible resource allocation. However, it is widely known that fairness and efficiency are usually not compatible with truthfulness. In this work, the authors studied a relaxed notion of truthfulness, namely, non-obvious manipulability (NOM). Fortunately, under the relaxed notion, the negative results do not hold anymore.

Originally, truthfulness means it is each agent’s dominant strategy to report the true values no matter what values will be reported by the others. The related notion NOM only requires that reporting the true values yields a (weakly) higher utility than lying in either the best and worst-case scenarios.

The fairness notion concerned in this work is EF1 (envy-free up to one good), a popular relaxation of EF (envy-freeness). The efficiency notions concerned include utilitarian welfare maximization, egalitarian welfare maximization, and Nash welfare maximization.

For fairness, under the new notion, the authors first show that the Round-Robin is actually NOM, which significantly separates truthfulness and NOM. Therefore, NOM and EF1 are compatible. Actually, using the algorithm by Aziz, WINE 2020, we can have a stronger result by achieving ex ante EF, ex post EF1 and NOM simultaneously.

For efficiency, (a little bit unexpected to me since the requirement of NOM is weak to me), the authors proved that for any number of agents, both egalitarian and Nash welfare maximizations are not compatible with NOM. For utilitarian welfare maximization, however, although it is still not compatible with NOM with two agents, they are compatible with more than two agents.

Finally, the authors investigated the compatibility among NOM, EF1, and Pareto efficiency. To answer this question, the authors provided a black box reduction from any algorithm that outputs (clean and non-wasteful) EF1 allocations to a new mechanism that not only ensures EF1 but is also NOM. Moreover, the reduction preserves the property of Pareto efficiency, which implies that NOM, EF1, and Pareto efficiency can be satisfied together, via existing results in the literature.


**Questions:**

For the reduction part, if all the valuations are strictly positive, the reduction actually does not need to do any change. I am not sure if this can be a good warm-up to show at the start of this section so that the readers can understand why we need to design the reduction like this.

The word “truthfulness” is used throughout the paper except related work, in which strategyproofness is sometimes used. I guess the authors want to be consistent with the referred papers, but I am not sure if the readers will get confused.

Have you ever defined what “PO” is short for?


**Limitations:**

N.A.

**Strengths And Weaknesses:**

Strengths

I agree with the authors that NOM might be a better notion for non-manipulability since truthfulness is hard to achieve together with fairness and efficiency. Thus, I think this work may be a good initiating work.

The authors also provide a relatively complete picture for NOM.


Weaknesses
The requirement of NOM is still a bit weak to me (although it is still not compatible with some efficiency criteria). It will be better if the authors can justify why only the best and the worst cases are particularly interesting.

The proofs are not technically hard.

The writing in general is good but can be further improved.


Typos:

Line 130 M should be non-italic in \Pi_n(M).
Line 214 l does not dependent of her report
Line 273 argmin should be a function
Line 274 also report her
Line 275 she get
Line 277 I do not see why Nash social welfare maximization is the most popular objective.
Line 329 each of these two subsets are
Line 351 some useful notation
Line 417 establish Theorem 5
Line 418 that that
“with respect to” is sometimes abbreviated and sometimes not; better to be consistent

---

> ### Author Response · Authors · 2022-07-31
> **Response to Reviewer Hf6D**
>
> Thank you for the thoughtful review and questions. Please see our response to all reviewers for your comment about the weakness of NOM as a definition.
>
> - “For the reduction part, if all the valuations are strictly positive, the reduction actually does not need to do any change. I am not sure if this can be a good warm-up to show at the start of this section so that the readers can understand why we need to design the reduction like this.”
>
> This is indeed the case for the “mechanics” (i.e. the code) of the reduction. However, for the NOM guarantee to go through, an agent must take into account the possibility of others reporting zero values when considering her best/worst-case outcomes. Specifically, we use the possibility of zero values in two places: (a) in the proof of Thm 6 (and specifically in lines 413-414), and (b) in the proof of Lemma 3 (lines 795-836). So, even if both the true and reported valuations are strictly positive, the reduction might be simpler, but the proof breaks down. For this reason we felt like this alternative presentation (presenting the case of the strictly positive values first) might mislead the reader into thinking that “if the valuation space is such that all values are always positive, then every EF1 + PO algorithm is also NOM”, which is not what we prove here.
>
> - “The word “truthfulness” is used throughout the paper except related work, in which strategyproofness is sometimes used. I guess the authors want to be consistent with the referred papers, but I am not sure if the readers will get confused.”
>
> We’d be happy to update all instances of “strategyproofness” with “truthfulness” (or add a footnote to explain that both terms refer to the same property).
>
> - “Have you ever defined what “PO” is short for?”
>
> PO stands for “Pareto optimal” or “Pareto efficient” (see line 164 in page 4). We will clarify this in other places as well, so it’s easier to track down for the reader.

---

### Author Response · Authors · 2022-07-31
**Response to all reviewers**

We would like to thank all reviewers for their constructive feedback. We will incorporate the valuable suggestions from all reviewers in the final version of this paper.

Reviewers Hf6D, iRAr and FBTg comment that NOM feels like a weak requirement and that it should be further motivated. We would like to first note that in the study of the allocation of indivisible items, a dominant research thread in fair division, most works point out that requiring truthfulness is very demanding (e.g., only dictatorships satisfy truthfulness + efficiency, while no mechanism is truthful and always EF1), and proceed to completely ignore the possibility of misreports. A series of recent papers that don’t fall in this category require instead a domain restriction (e.g. binary values). We view our work not as advocating NOM as the ultimate guarantee one should aim for, but as initiating the exploration of formal guarantees between “agents are always honest” and “agents are perfectly rational and all-knowing expected utility maximizers.” So, NOM for us is a relaxation of truthfulness as much as it is a strengthening of “absolute honesty.”

Regarding general motivation for NOM, NOM protects against agents that consider the best and worst case outcomes under different reports. This is a much more realistic assumption compared to truthfulness, which requires an agent to argue about *all* possible scenarios (which is clearly impossible for a real person to do). And, even though a real person might be more sophisticated than simply considering these two extremes, NOM does manage to separate non-truthful mechanisms in terms of how manipulable they are in a way consistent with evidence from practice. For example, the Boston mechanism (which has been observed to be manipulable in practice) is also obviously manipulable, while the deferred acceptance algorithm (which is widely believed to be fairly robust against most manipulations, since manipulating it requires a detailed understanding of others’ preferences) does satisfy NOM; see [Troyan and Morrill, 2020].

A technical benefit of arguing only about the min and max outcome is that conclusions are not tied to any distributional assumptions on preferences. Finally, on another technical note, even though NOM sounds weak, as we show, it is not so weak that NOM+X is always possible (e.g. NOM + MNW and NOM + egalitarian welfare are not possible), so there is a clear separation between “no incentives,” NOM, and standard truthfulness.

Below we respond to each reviewer individually.

---

### Meta-Review · Area_Chair_orpm · 2022-08-26

**Recommendation:** Accept
**Confidence:** Certain

**Metareview:**

Reviewers agreed that this paper explored a natural and interesting strategic aspect of fair division (non-obvious manipulability). This helped escape classical impossibility results in fair division. Minor concerns were raised about the practical significance of NOM, but overall the sentiment was quite positive.

**Award:**

No

---

### Decision · Program_Chairs · 2022-09-14

Accept